# Phosphorylation of mouse intestinal basolateral amino acid uniporter LAT4 is controlled by food-entrained diurnal rhythm and dietary proteins

**Lalita Oparija-Rogenmozere**[1¤a], **Anuradha Rajendran**[1¤b], **Nadège Poncet**[1], **Simone M. R. Camargo**[1], **François Verrey**[1,2]*

**1** Institute of Physiology and Zurich Center for Integrative Human Physiology (ZIHP), University of Zurich, Zurich, Switzerland, **2** NCCR Kidney.CH, Zurich, Switzerland

¤a Current address: Department of Anatomy and Neuroscience, University of Melbourne, Melbourne, Australia
¤b Current address: Beth Israel Deaconess Medical Center, Harvard Medical School, Boston, Massachusetts, United States of America
* francois.verrey@uzh.ch

**Data Availability Statement:** Relevant data are shown in the paper in forms of figures, either with

## Abstract

Adaptive regulation of epithelial transporters to nutrient intake is essential to decrease energy costs of their synthesis and maintenance, however such regulation is understudied. Previously we demonstrated that the transport function of the basolateral amino acid uniporter LAT4 (Slc43a2) is increased by dephosphorylation of serine 274 (S274) and nearly abolished by dephosphorylation of serine 297 (S297) when expressed in *Xenopus* oocytes. Phosphorylation changes in the jejunum of food-entrained mice suggested an increase in LAT4 transport function during food expectation. Thus, we investigated further how phosphorylation, expression and localization of mouse intestinal LAT4 respond to food-entrained diurnal rhythm and dietary protein content. In mice entrained with 18% protein diet, LAT4 mRNA was not submitted to diurnal regulation, unlike mRNAs of luminal symporters and antiporters. Only in duodenum, LAT4 protein expression increased during food intake. Concurrently, S274 phosphorylation was decreased in all three small intestinal segments, whereas S297 phosphorylation was increased only in jejunum. Interestingly, during food intake, S274 phosphorylation was nearly absent in ileum and accompanied by strong phosphorylation of mTORC1 target S6. Entraining mice with 8% protein diet provoked a shift in jejunal LAT4 localization from the cell surface to intracellular stores and increased S274 phosphorylation in both jejunum and ileum during food anticipation, suggesting decreased transport function. In contrast, 40% dietary protein content led to increased LAT4 expression in jejunum and its internalization in ileum. *Ex vivo* treatments of isolated intestinal villi fraction demonstrated that S274 phosphorylation was stimulated by protein kinase A. Rapamycin-sensitive insulin treatment and amino acids increased S297 phosphorylation, suggesting that the response to food intake might be regulated via the insulin-mTORC1 pathway. Ghrelin, an oscillating orexigenic hormone, did not affect phosphorylation of

individual data points shown or with explanation how measurements and analysis were done to calculate means and statistical errors. All primary data are documented and preserved at the Institute of Physiology at University of Zurich in accordance with the guidelines of the Swiss Academy of Medical Sciences (https://www.physiol.uzh.ch/en.html).

**Funding:** F.V.: Swiss National Science Foundation grant #31_166430/1. Funder website: http://www.snf.ch The funders had no role in study design, data collection and analysis, decision to publish, or preparation of the manuscript.

**Competing interests:** The authors have declared that no competing interests exist.

intestinal LAT4. Overall, we show that phosphorylation, expression and localization of intestinal mouse LAT4 responds to diurnal and dietary stimuli in location-specific manner.

## Introduction

The small intestine is the largest part of the intestinal barrier between the external environment and the body interior [1]. It can be divided into three major parts: duodenum, followed by jejunum and ileum; with its main role being nutrient digestion and absorption [2]. The intestinal mucosa is lined by epithelial cells, called enterocytes, that express several apical and basolateral solute carriers (SLCs) which absorb luminal nutrients, such as amino acids, and deliver them into the bloodstream [3–5].

Porcine intestinal transcriptome analysis has revealed that most solute carriers responsible for nutrient uptake show a region-specific expression along the longitudinal axis and that duodenum expresses the highest mRNA levels for many of these SLC family members [6, 7]. However, these studies, together with more targeted ones in other species, show that many of the mRNAs encoding nutrient transporters, such as the glucose transporter SGLT1 (Slc5a1) and most epithelial amino acid transporters are expressed at highest levels in jejunum and/or ileum [7, 8].

Epithelial cells in each intestinal segment are further ordered into crypts and villi. The crypts are responsible for self-renewal of the epithelia as each crypt contains four to six pluripotent epithelial stem cells. The newly produced cells differentiate into one of various intestinal cell lineages and move upwards to the villus tip where they are eventually shed off within approximately 5 days. The majority of cells along the villi are enterocytes, which are differentiated for digestive, absorptive and metabolic function [1, 2, 9]. A recent study looking at the mouse transcriptome along the villus axis has revealed that about 80% of the genes expressed in enterocytes have a specific zonal localization and that therefore villi can be divided into tip, middle and bottom functional zones. Regarding nutrient uptake, carbohydrate and amino acid transporters (AATs, 27 out of 41 tested) were found to be predominantly expressed in the middle of the villi, including the basolateral amino acid uniporter LAT4 (Slc43a2) [10]. We have previously shown that specifically in this middle zone of the villi LAT4 undergoes most of the serine 274 (S274) dephosphorylation during food anticipation. This posttranslational modification, according to our observations in *Xenopus* oocytes, leads to an increase in LAT4 affinity and transport function [11]. These data reveal a complex functional zonation along both the longitudinal and the crypt-villus axes of the small intestine. This is an important aspect that needs to be considered when investigating intestinal epithelial transport, since most recent studies have focused only on a specific segment of the small intestine or on the small intestine as a whole, without segment separation.

Amino acid intake depends on the daily rhythm and the diet composition, such that the small intestine must adjust to variable concentrations of luminal peptides and amino acids. Thirty years ago, it was suggested that the benefits of amino acid transport should exceed the costs of transporter synthesis and maintenance, thus the transport capacity and/or apparent affinity should be adapted to the luminal substrate content. At that time, regulation was thought to happen mostly at the transcriptional and translational levels, however it was speculated that post-translational mechanisms might be involved as well, for example transporter endocytosis and/or degradation [12].

Unfortunately, current knowledge on adaptive signals and mechanisms of epithelial amino acid transport adaptation is almost as scarce as it was thirty years ago. So far there are only two non-epithelial AAT systems, system A and system y⁺, whose adaptive regulation has been investigated in more detail. Importantly, these studies have identified some potential regulatory mechanisms and signaling pathways that could be used also for epithelial transport regulation. Both system A (specifically SNAT2, Sls38a2) and system y⁺ (CAT-1, Slc7a1) display functional adaptation to their own substrates and regulation by gastrointestinal hormones such as insulin. Their regulation seems to involve the modulation of gene transcription and translation, yet there are some indications that also inactivation of synthesized transporters might take place, possibly by posttranslational modification [13–17].

Mammalian target of rapamycin complex 1 (mTORC1) also appears to be involved in AAT regulation. Activation of mTORC1 itself is mediated by the co-incident availability of nutrients, particularly of a high intracellular leucine concentration, and stimulation by growth factors such as insulin. Some of the main mTORC1 targets are ribosomal protein S6 and its kinase, their phosphorylation being a key indication of mTORC1 activity [18–20]. Stimulation of cultured muscle cells with leucine has been shown to induce transport via system A. This effect was inhibited by wortmannin (phosphatidylinositol 3-kinase inhibitor) and rapamycin, suggesting a regulation by insulin via mTORC1 signaling [21]. Studies of placental amino acid transport have revealed regulation via the insulin-mTORC1 pathway as well, with low insulin concentrations leading to downregulation of mTORC1 activity and low expression of multiple AATs, [22] specifically of systems A and L [23, 24].

Small intestinal cells also express multiple clock genes that generate a self-sustaining diurnal rhythm. The intestinal clock communicates with the light-entrained central clock in the brain, however the most powerful clock entrainment signal in the periphery is rhythmic food intake. Current findings in rodents suggest that in the small intestine and in the liver, food-entrainment signals dominate and overwrite the oscillations given by the central clock [25–27]. Previously we have summarized known AATs that show diurnal rhythmicity and demonstrated that also the uniporter LAT4 exhibits a food anticipatory response, specifically at the level of its phosphorylation that is modified at the start of the feeding time in food-entrained mice. Our functional experiments in *Xenopus* oocytes had revealed two important phosphorylation sites on LAT4: S274 and serine 297 (S297). Dephosphorylation of S274 positively impacted LAT4 function, whilst the dephosphorylation of serine S297 led to a decrease in LAT4 substrate affinity and transport rate, nearly abolishing its function. Food-entrained mice showed at the anticipated feeding time an increase in jejunal S297 phosphorylation and a decrease in phosphorylated S274 levels all along the small intestine, suggesting increased transport capacity [11].

Precise signaling pathways or effectors mediating the diurnal regulation of amino acid transport have not yet been described. The orexigenic hormone ghrelin was considered as a candidate effector, being known to peak in the plasma of humans and rodents shortly before the anticipated food intake and to rapidly decrease afterwards [28–30]. In addition, a single ghrelin injection was shown to cause an acute increase in the appetite and food intake in mice and ghrelin secreting cells in the stomach have been suggested to be the peripheral food-entrainable oscillators causing food anticipatory activity [31, 32]. Yet, the majority of the studies with ghrelin or ghrelin receptor knockout mice still detected food anticipatory behaviors, questioning the hypothesis that ghrelin is the major food-entrainable oscillator in periphery [33]. It has also been suggested that another ligand might bind to ghrelin receptors to induce the food anticipatory response [34]. In addition, it is unclear whether ghrelin receptors (growth hormone secretagogue receptors, GHS-R1α) are expressed in the small intestine at all —no mRNA expression has been detected in human intestinal samples [35] and very low and

low mRNA expression has been detected in the small intestine of guinea pig and rat, respectively [36]. Overall, there is a lack of solid information on ghrelin receptor protein levels and localization in the small intestine of mammals. To our knowledge, there are also no reported studies investigating the role of ghrelin in diurnal regulation of intestinal nutrient transporters.

In this study, we tested the hypothesis that the expression and/or phosphorylation of the intestinal amino acid uniporter LAT4 is subjected to adaptive regulation in the context of the food-entrained diurnal rhythm and in response to the dietary protein content. Thus, we entrained wild-type mice by time restricted feeding using standard (18%) as well as low (8%) and high (40%) protein diets to evaluate their impact on LAT4 expression, phosphorylation and localization. Our initial tests suggested that the impact might vary along the intestine, so we investigated in depth whether there are indeed regulatory differences between duodenum, jejunum and ileum.

To identify possible regulatory pathways that mediate the (de-)phosphorylation of LAT4, we investigated the effect of some likely candidates by using a new experimental model, namely the *ex vivo* treatment of isolated murine intestinal villi epithelia. Based on predicted kinase motifs [37], we tested the possible involvement of protein kinase A, of insulin (via mTORC1 pathway), and of free amino acids on the regulation of LAT4 phosphorylation. We also tested whether the direct action of ghrelin might affect S274 or S297 phosphorylation of LAT4 by injecting this hormone *in vivo* to food-entrained WT mice during their active and their resting phase.

## Materials and methods

### Ethical approval

All experimental procedures and handling involving mice were approved by the Cantonal Veterinary Office of Zurich (references #ZH075/15; #ZH228/17 and #ZH206/18) and performed in accordance with Swiss Animal Welfare laws.

### Origin and source of the mice strains used

Mice from three different backgrounds were used as wild type (WT) mice during the study. To assess the food-entrained diurnal regulation, to perform *in vivo* treatments with ghrelin and *ex vivo* treatments of isolated intestinal villi, 8–10 weeks old C57BL/6J (IMSR Cat# JAX:000664, RRID:IMSR_JAX:000664, Charles River Laboratories, Inc., Wilmington, MA, USA) male mice were used. One mouse from the diurnal regulation experiment died shortly after arrival from the commercial breeding facility and thus was excluded from the study.

To perform mRNA tests and to assess dietary regulation under food entrainment with 8% and 18% protein diets, we used 8 weeks old LAT4flx/flxROSACreERT2- (strain background described in [11]), male and female mice. Intestinal fractions prepared from tamoxifen-induced LAT4flx/flxROSACreERT2+ mice were used as negative controls for antibodies for Western blotting (further referenced as LAT4 conditional knockout). Induction was done by intraperitoneal injection of 3 mg tamoxifen (vehicle: corn oil with 5% EtOH) per 25g body weight for 5 consecutive days.

For food entrainment with 18% and 40% protein diets, custom made LAT4flx/flx mice (C57Bl/6N background, PolyGene AG, Rumlang, Switzerland) were bred with tamoxifen-inducible VillinCreERT2- mice (B6N.Cg-Tg(Vil1-cre/ERT2)23Syr/J) (Cat# JAX:020282, RRID:IMSR_JAX:020282, The Jackson Laboratory, Bar Harbor, ME, USA) in the Laboratory Animal Services Center (LASC) facilities at the University of Zurich. We did not observe any differences in intestinal LAT4 protein expression or phosphorylation levels between

**Table 1. Experimental mice strains and total numbers used.**

| Experiment/-s | Figure/-s | Strain used | Sex | n |
|---|---|---|---|---|
| mRNA expression test, dietary regulation: 8% and 18% protein | Figs 1B and 8–11 | LAT4$^{flx/flx}$ROSACreERT2$^-$ | Males and females | 19 |
| 8% and 18% protein diets; *ex vivo* treatments of isolated intestinal villi | Negative controls for Figs 2, 3, 12 and 13 and S1 Fig | LAT4$^{flx/flx}$ROSACreERT2$^+$ | Males and females | 9 |
| Dietary regulation: 18% and 40% protein diets | Figs 8–11 | LAT4$^{flx/flx}$VillinCreERT2$^-$ | Males and females | 17 |
| Diurnal regulation | Figs 2–7 | C57BL/6J | Males | 65 |
| Ghrelin or saline i.p. injections | S1 Fig | C57BL/6J | Males | 24 |
| *Ex vivo* treatments of isolated intestinal villi | Figs 12 and 13 | C57BL/6J | Males | 30 |

ROSACreERT2$^-$ and VillinCreERT2$^-$ mouse models, thus data from both strains were pooled and analyzed together. All mice used for experiments were aged 8 to 10 weeks and were within the weight range of 18 to 35 grams. Samples collected during the study were processed in randomized and, where possible, in blinded manner. Detailed information on strains and animal numbers per experiment are shown in Table 1.

## Animal housing and food entrainment for ghrelin, diurnal and dietary regulation experiments

Eight to ten weeks old mice from relevant strains (Table 1), were housed in standard laboratory conditions and fed 18% protein diet (Kliba Nafag, Kaiseraugst, Switzerland) *ad libitum* for approximately one week after their arrival from commercial or breeding facility. Afterwards mice were transferred to climate chambers with inverted 12 h dark/light cycle, the dark (active) period starting at 9:00 (ZT12). Temperature in the climate chambers was kept constant at 22˚C. After one week adjustment to the inverted rhythm in the climate chamber, time restricted feeding (TRF) was started with the chow diet given from ZT12 to ZT20 to synchronize the intestinal clock. Mice were kept on TRF for at least 14 days with water provided *ad libitum*. For mice used in the food-entrained diurnal regulation experiments (Figs 1B and 2–7) or for ghrelin *in vivo* injections (S1 Fig) TRF was done using the same 18% protein diet. For mice used in dietary regulation experiments (Figs 8–11) isocaloric diets (maintained with carbohydrate substitution) with 8%, 18% or 40% protein (casein) content were used for food entrainment (all diets from Kliba Nafag, Kaiseraugst, Switzerland), with 18% protein diet serving as a control diet. All mice used in this study were anesthetized with Attane™ Isoflurane (Cat# NDC 66794-014-10, multiple lots used, Primal Critical Care Inc., Bethlehem, PA, USA) in oxygenated chamber and, after falling unconscious, euthanized by cervical dislocation. Mice used for the mRNA expression tests and ghrelin *in vivo* injections were euthanized at ZT0 and ZT12 (Fig 1B and S1 Fig); mice used in dietary regulation tests—only at ZT12 (Figs 8–11). To assess the diurnal regulation, mice were euthanized at ZT0, ZT4, ZT8, ZT12, ZT16 and ZT20 (Figs 2–7).

In all experiments, mice euthanasia at ZT12 was done before food addition and at ZT20 before food removal. For mRNA expression and diurnal regulation experiments, ZT0 (lights on, resting phase starts) was used as a control timepoint.

## *In vivo* ghrelin injections

Food-entrained C57BL/6J male mice were randomly assigned to one of the injection time points: ZT12 (start of the active phase, food expected, but not yet added) and ZT0 (start of the resting phase, 4 hours after food removal). Mice received a single intraperitoneal (i.p.)

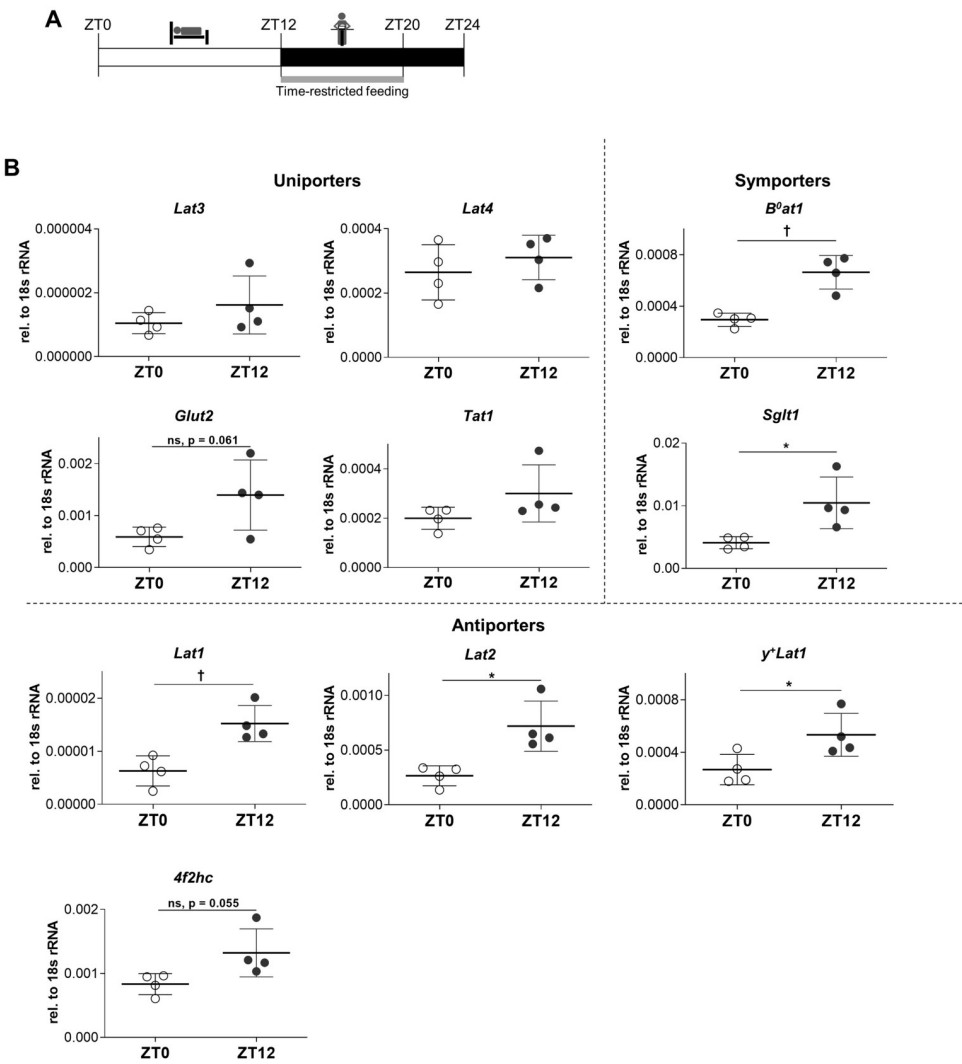

**Fig 1. mRNAs of nutrient uniporters in mice duodenum show little or no food-entrained diurnal regulation, in contrast to symporters and antiporters. A**: Food-entrainment regimen used for all mice experiments. ZT0: start of the resting period; ZT12: start of the active period. Mice fed 18% protein diet only from ZT12 to ZT20 for at least 14 consecutive days. Euthanasia at ZT12 was done before the feeding starts. **B**: mRNA expression of different nutrient transporters in duodenum measured by real time PCR. Statistical analysis performed using Student's unpaired t test, *p<0.05; †p<0.01. Mean (SD), n = 4 mice from a single experiment.

injection of 100 μl physiological saline, either with or without 10 μg of ghrelin (rat ghrelin, Cat#1465, Lot# 27B, Tocris Bioscience, Bristol, UK) (n = 6 saline + 6 ghrelin injected mice per time point). Fifteen minutes after the injection all mice were euthanized as described in the previous section. Duodenum and jejunum were collected for intestinal villi fraction isolation as described in the corresponding section below. Blood plasma was collected and processed for quantitative determination of growth hormone concentration, using m/rGH ELISA Kit (Cat# E023, Lot# 111217, Mediagnost, Reutlingen, Germany), according to the manufacturer's instructions. The absorbance was measured with a plate reader Infinite M200 PRO (Tecan, Tecan Life Sciences, RRID:SCR_016771, Maennerdorf, Switzerland) at 450 nm with 590 nm as reference wavelength.

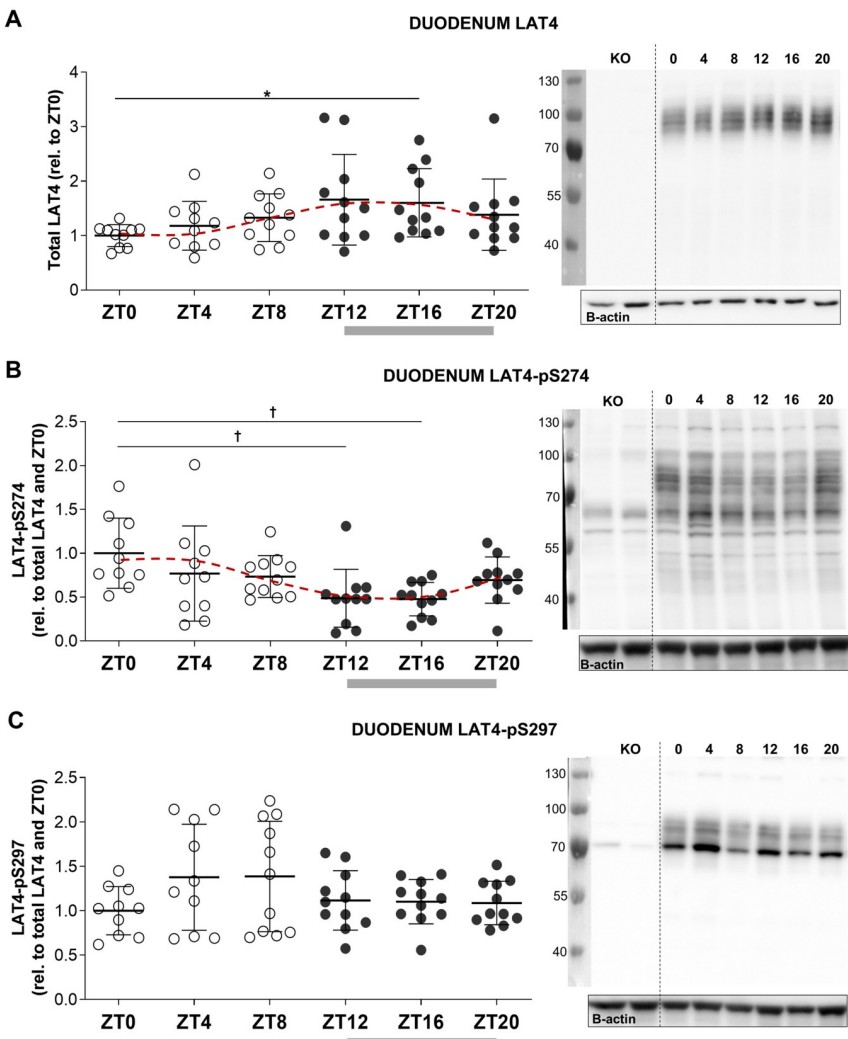

**Fig 2. LAT4 protein expression and phosphorylation on S274 show diurnal rhythmicity in duodenum of food-entrained wild type mice.** Total membrane lysates from duodenum of wild type (WT) and conditional LAT4 knockout (KO) mice analyzed by Western blot. Grey bar shows feeding period, red dotted line shows fitted cosine wave. **A**: LAT4 protein expression in duodenum. **B**: LAT4 phosphorylation on S274. **C**: LAT4 phosphorylation on S297. To quantify, all values were normalized to beta actin. Further normalization to LAT4 ratio was done for phospho-specific antibodies. Quantification with means ± SD is shown in the left panel, representative blots in the right panel. Statistical analysis performed with One-way ANOVA using Tukey's multiple comparison test, *p<0.05; †p<0.01. Mean (SD), n = 10–11 mice per time point from two separate experiments.

## Preparation of isolated intestinal fractions

Isolated intestinal villi fractions were prepared for experiments of food-entrained diurnal regulation, *in vivo* ghrelin injection and all *ex vivo* treatments. The protocol was adapted from [38, 39]. After euthanasia, the small intestine was removed and placed on a plain, clean surface at room temperature (RT) and cut open longitudinally. A glass coverslip was used to mechanically remove the intestinal contents and mucus by gentle scraping at a 45° angle, along the intestinal surface. Afterwards, the intestine was cut into 1–2 cm pieces and placed into 50 ml conical tube with 20 ml ice-cold PBS (Cat#10010023, multiple lots used, Gibco via Thermo Fischer Scientific, Waltham, MA, USA) containing cOmplete Mini EDTA-free Protease

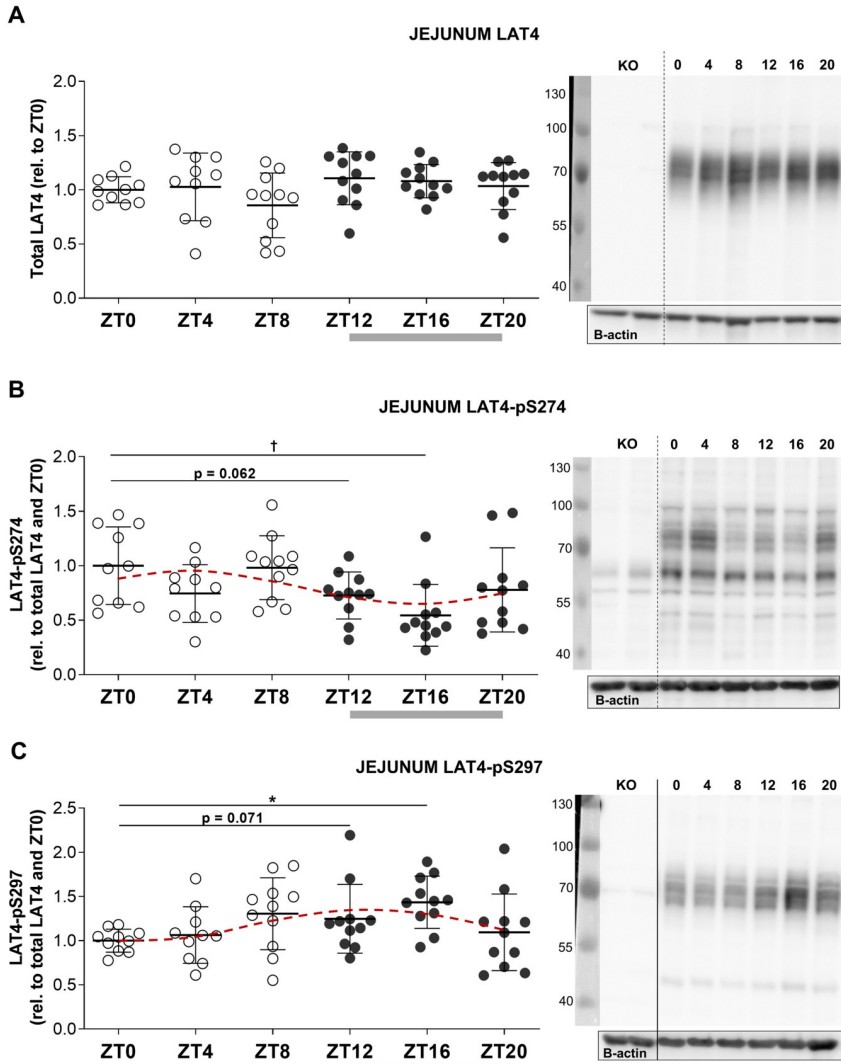

**Fig 3. Only phosphorylation of LAT4 exhibits diurnal rhythmicity in jejunum of food-entrained wild type mice.**
Grey bar shows feeding period, red dotted line shows fitted cosine wave. **A**: LAT4 protein expression in jejunum. **B**:
LAT4 phosphorylation on S274. **C**: LAT4 phosphorylation on S297. **A to C**: Total membrane lysates from jejunum of
wild type (WT) and conditional LAT4 knockout (KO) mice analyzed by Western blot. Uninterrupted line between KO
and other samples in the blot image indicates splicing—lanes from the same blot were rearranged to match the sample
sequence of the blots shown in 3A and 3B. To quantify, all values were normalized to beta actin. Further normalization
to LAT4 ratio was done for phospho-specific antibodies. Quantification is shown in the left panel, representative blots
in the right panel. Statistical analysis was performed with One-way ANOVA using Tukey's multiple comparison test,
*p<0.05; †p<0.01. Mean (SD), n = 10–11 mice per time point from two separate experiments.

inhibitor cocktail tablets (all experiments, 1 tablet per 7 ml of PBS) (Cat# 4693159001) and
Phosphatase inhibitor cocktail tablets PhosSTOP (diurnal and ghrelin experiments only, 1 tab-
let for 10 ml of PBS) (Cat# 04906837001) (both: multiple lots used, from Roche via Sigma-
Aldrich, Buchs, Switzerland). For assessment of food-entrained diurnal regulation and regula-
tion via ghrelin, further processing of duodenum and jejunum was done separately. Ileum was
processed only for immunofluorescence as described below. For all *ex vivo* treatment experi-
ments, the whole small intestine was processed as a whole due to the high number of villi
needed.

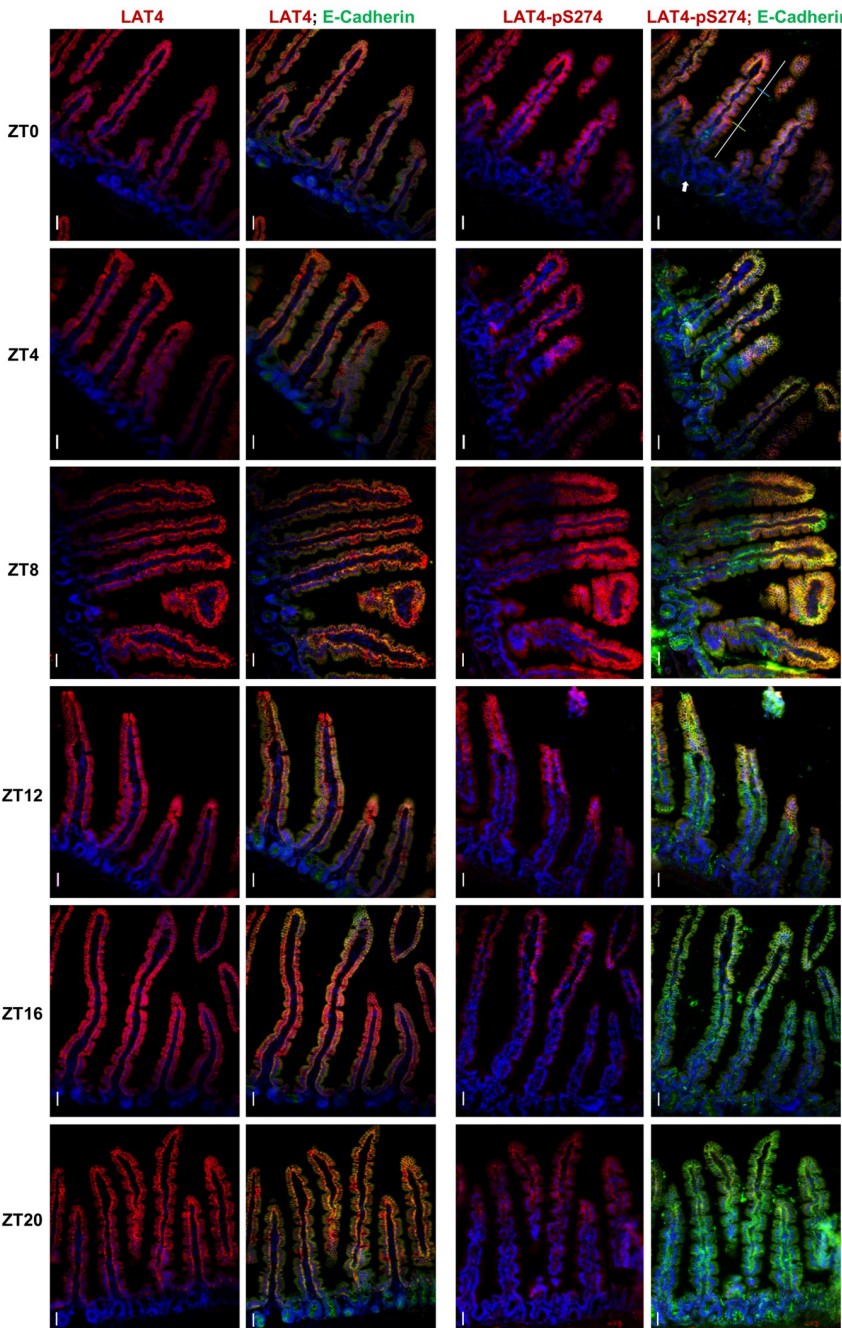

**Fig 4. Phosphorylation on S274 in jejunum exhibits patchy localization and decreases during food anticipation and intake.** Image at the upper right corner: white line marks the full length of villus, blue line demonstrates our applied border between the tip and middle part, and green line—border between middle and base part. Arrowhead marks crypt, where neither LAT4 nor pS274 staining is present. Red: LAT4 or pS274, green: E-Cadherin, blue: DAPI. Scale bar: 30 μm.

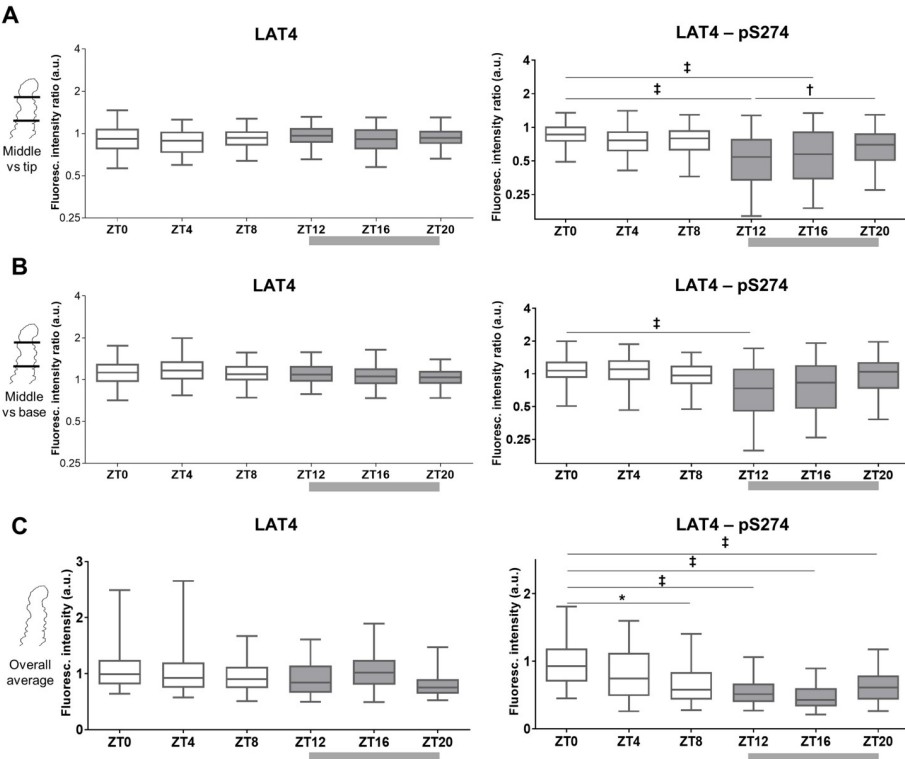

**Fig 5. S274 phosphorylation in jejunum is decreased mostly at the middle part of the villus. A**: Fluorescence ratio between the middle part and tip of jejunal villi. Logarithmic scale used. 1: same fluorescence intensity on middle part and tip, >1: fluorescence intensity is increased at the middle part, <1: intensity lower at middle part. **B**: Fluorescence ratio between the middle part and the base of jejunal villi. **C**: Quantitation of fluorescence intensity of jejunal villi. **A to C**: villi were stained with antibodies against LAT4 (left panels) or pS274 (right panels), and membrane marker E-cadherin. Fluorescence intensity determined with ImageJ 1x software and related to intensity of E-cadherin. Box at 25th and 75th percentile, line at median and whiskers at 5th to 95th percentile, n = 250–300 villi from 10–11 mice per time point from two separate experiments. Statistical analysis: One-way ANOVA using Tukey's multiple comparison test, $^*p<0.05$; $^\dagger p<0.01$; $^\ddagger p<0.001$.

Afterwards intestinal pieces were washed three times with 20 ml of ice-cold PBS by vigorous shaking for 10 s. After washing, they were incubated in PBS containing 2 mM EDTA and incubated for 30 min at 4°C with slow rotation. Then the solution was removed, 20 ml of PBS were added and tubes gently inverted 5 times. The supernatant containing the detached villi was inspected and collected. Another 20 ml of PBS were added, and the tubes vigorously inverted 15 times. Again, the supernatant containing detached villi was inspected and collected. The suspension was centrifuged at 1500 rpm at 4°C for 5 minutes to sediment the villi fraction. Afterwards the supernatant was removed, and the pellet resuspended in PBS. To perform the *ex vivo* treatments, the resuspension was evenly split and distributed in 1.5 ml tubes; the resuspension of samples from diurnal regulation and ghrelin experiments was kept in the same tube. Then it was centrifuged at 1000 rpm at 4°C for 3 minutes and the supernatant containing single cells and cell debris was removed. Afterwards the villi fractions were processed for *ex vivo* treatments as described in the corresponding section below. Pellets from diurnal regulation and ghrelin experiments were resuspended in 200 μl mannitol resuspension buffer (200 mM D-Mannitol, 80 mM HEPES, 41 mM KOH, pH 7.5) with supplemented Protease inhibitor (Cat# P8340, multiple lots used, Sigma-Aldrich, Buchs, Switzerland) and Phosphatase

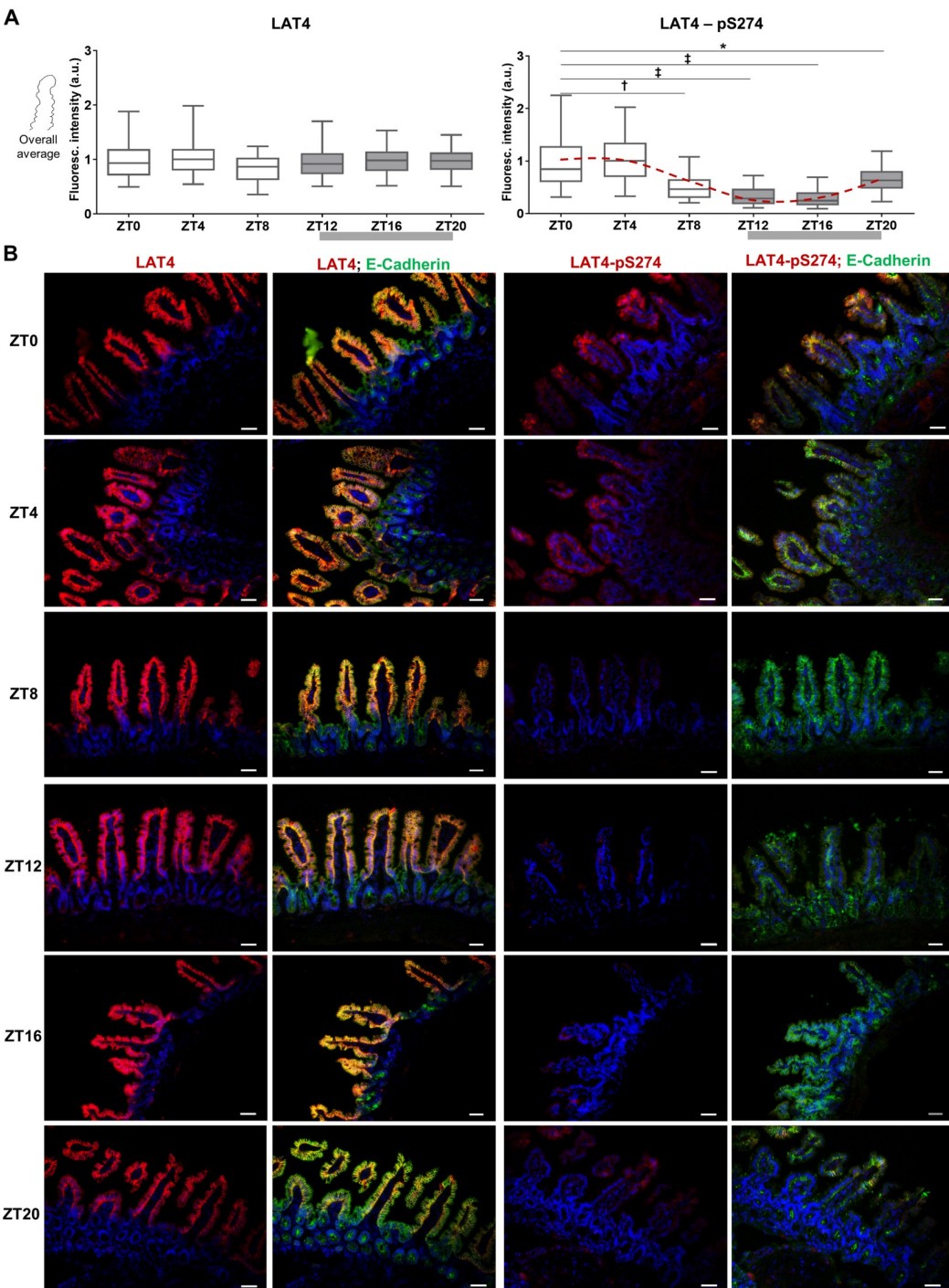

**Fig 6. Phosphorylation of LAT4 on S274 is strongly decreased in ileum during food anticipation and intake.** Grey bar shows feeding period, red dotted line shows fitted cosine wave. **A**: Quantitation of fluorescence intensity on ileal villi of food-entrained wild type mice stained with antibodies against LAT4 (left panel) or pS274 (right panel), and membrane marker E-cadherin. Fluorescence intensity determined with ImageJ 1x software was normalized to the intensity of E-cadherin. Median shown, n = 300–350 villi from 10–11 mice per time point from two separate experiments. Statistical analysis: One-way ANOVA with Tukey's multiple comparison test, *p<0.05; †p<0.01; ‡p<0.001. **B**: LAT4 expression and phosphorylation on S274 in ileum. Red: LAT4 or pS274, green: E-Cadherin, blue: DAPI. Scale bar: 30 μm.

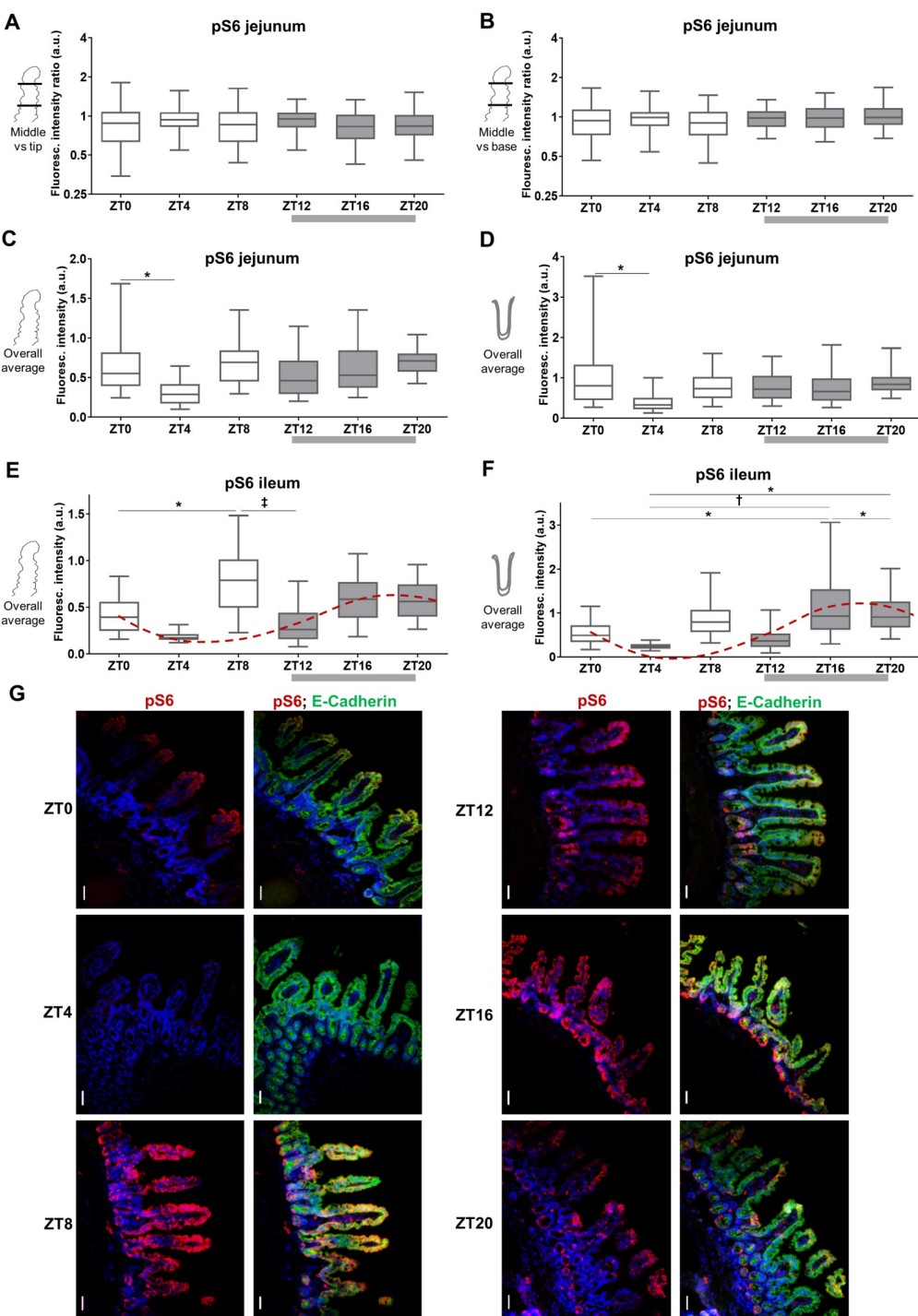

**Fig 7. Stronger response to food-entrainment of mTOR activity in ileum than jejunum.** Grey bar shows feeding period, red dotted line shows fitted cosine wave (without the ZT8 values). **A**: Fluorescence ratio between the middle part and tip of jejunal villi. Logarithmic scale used. 1: same fluorescence intensity on middle part and tip, >1: fluorescence intensity is increased at the middle part, <1: intensity lower at middle part. **B**: Fluorescence ratio between the middle part and the base of jejunal villi. **C**: Quantitation of fluorescence intensity of jejunal villi. **D**: Quantitation of fluorescence intensity of jejunal crypts. **E**: Quantitation of fluorescence intensity of ileal villi. **F**: Quantitation of fluorescence intensity of ileal crypts. **A to F**: villi were stained with antibody against pS6, and membrane marker E-cadherin. Fluorescence intensity determined with ImageJ 1x software and normalized to intensity of E-cadherin. Median shown, n = 110–190 (jejunal villi); n = 160–280 (jejunal crypts), n = 140–220 (ileal villi); n = 140–250 (ileal

crypts) from 10–11 mice per time point from two separate experiments. Statistical analysis: One-way ANOVA with Tukey's multiple comparison test, *p<0.05, †p<0.01, ‡p<0.001. **G**: Phosphorylation on S6 in mice ileum. Red: pS6, green: E-Cadherin, blue: DAPI. Scale bar: 30 μm.

inhibitor (Cat# 100567, Lot# 11117006, Active Motif, La Hulpe, Belgium) cocktails and lysed using a tip sonicator (Labsonic 1510, Bender&Hobein, Zurich, Switzerland) for 2 x 5 s. The lysates were centrifuged at 2000 g for 15 min at 4˚C to sediment cellular debris, the supernatants were then collected and ultra-centrifuged at 41 000 rpm for 1h at 4˚C (Rotor RP45A, Sorvall, ThermoFisher Scientific, Waltham, MA, USA). Afterwards the supernatant was discarded and the pellet containing total membranes was resuspended in 30 μl of mannitol resuspension buffer using the tip sonicator. Resuspended samples were snap frozen in liquid nitrogen and stored at -80˚C.

### *Ex vivo* treatments of isolated intestinal villi fractions

After the separation of the intestinal villi from single cells and cell debris, their pellets were resuspended in 0.5 ml of either phosphate buffer (for insulin, rapamycin and amino acid treatments) (118 mM NaCl, 4.7 mM KCl, 1.2 mM $KH_2PO_4$, 1.8 mM $CaCl_2$, 1.44 $MgSO_4x7H_2O$, 2mM Glucose, 10 mM $NaHCO_3$, 10 mM HEPES) or cell culture media (for forskolin, H89 and LB100 treatments) (custom formulation: DMEM/F12 without any amino acids, L-Glutamine and bovine serum; with 15 mM HEPES and 1.2 g $L^{-1}$ $NaNHCO_3$; Cat# P04-41507, Lot# 5821217, PAN Biotech, Aidenbach, Germany). Two different incubation solutions were chosen based on our preliminary testing results—cell culture media led to longer cell viability and slower LAT4 degradation, however due to the higher glucose concentration it was considered as not suitable for the treatments with insulin or amino acids.

The specific effectors were chosen based on the predicted kinase motifs for LAT4 S274 and S297 sites, shown in the PhosphoMotif Finder, an online tool from the Human Protein Reference Database [37] (available at: http://www.hprd.org/PhosphoMotif_finder) and the PhosphoNet, a human phosphosite KnowledgeBase developed by Kinexus (available at: http://www.phosphonet.ca/).

To activate protein kinase A (PKA), 25 μM forskolin (Cat#F3917, Lot#SLBS3581V, Sigma-Aldrich, Buchs, Switzerland) was used. To inhibit PKA, we pre-incubated the villi for 5 min with 30 μM H89 dihydrochloride hydrate (Cat#B1427, Lot#037M4702V, Sigma-Aldrich, Buchs, Switzerland) before adding forskolin. To inhibit protein phosphatase 2A (PP2A), we used its specific inhibitor LB100 (Cat# S7537, Lot# S753701, Selleckchem, Munich, Germany) at 10 μM concentration. For mock incubation, samples were treated with DMSO (solvent for forskolin) and/or $H_2O$ (solvent for H89 and LB100).

To activate insulin signaling, we used human recombinant insulin (Cat# 12643, Lot# SLBR9404V, Sigma-Aldrich, Buchs, Switzerland) at 10 nM concentration. To inhibit mTORC1 signaling, villi were pre-incubated with 100 nM rapamycin (Cat# R0395, Lot# 077M4007V, Sigma-Aldrich, Buchs, Switzerland) for 5 minutes before insulin was added. For amino acid treatments, we used an amino acid mix of 5 x the physiological plasma concentration (physiological concentrations and composition as shown in [40]). Short preincubation times were also chosen based on our preliminary testing—LAT4 showed time-dependent degradation in samples, thus long treatments (>20 min) were unfavorable.

All *ex vivo* treatments were done for 10 min at 37˚C with rotation at 225 $min^{-1}$. After the incubation, samples were immediately centrifuged at 2000 rpm at 4˚C for 5 min to pellet the fractions. The supernatant was discarded, and pellets were resuspended in 60 μl mannitol

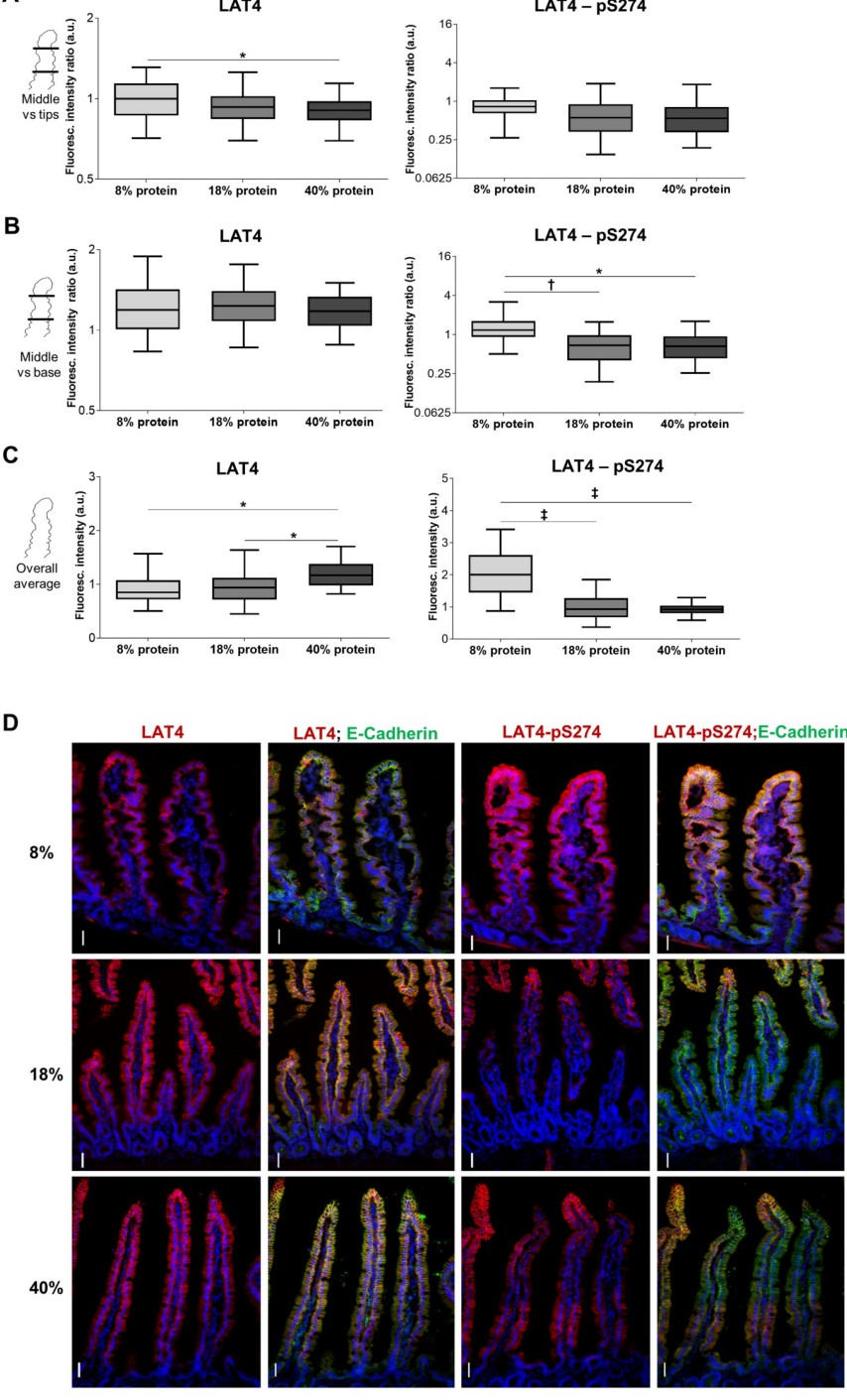

**Fig 8. Low protein diet leads to increase in phosphorylation on S274 in jejunum of food-entrained mice during food anticipation.** Mice were food-entrained (TRF) with low (8%), standard (18%) or high protein diet (40%) for at least 14 consecutive days and euthanized at ZT12. **A**: Fluorescence ratio between the middle part and tip of jejunal villi. Logarithmic scale used. 1: same fluorescence intensity on middle part and tip, >1: fluorescence intensity is increased at the middle part, <1: intensity lower at middle part. **B**: Fluorescence ratio between the middle part and the base of jejunal villi. **C**: Quantitation of fluorescence intensity of jejunal villi. **A to C**: villi were stained with antibodies against LAT4 (left panels) or pS274 (right panels), and membrane marker E-cadherin. Fluorescence intensity was determined with ImageJ 1x software and normalized to intensity of E-cadherin. Median shown, n = 150–300 villi from 4–5 mice per diet from two separate experiments. Statistical analysis: One-way ANOVA with Tukey's multiple comparison test,

*p<0.05; †p<0.01; ‡p<0.001. **D**: LAT4 expression (left panels) and phosphorylation on S274 (right panels) in mice jejunum. Red: LAT4 or pS274, green: E-Cadherin, blue: DAPI. Scale bar: 30 μm.

resuspension buffer with supplemented Protease inhibitor and Phosphatase inhibitor cocktails and lysed using a tip sonicator for 2 x 5 s. Afterwards total lysates were snap frozen in liquid nitrogen and stored at -80˚C.

## Antibodies used for Western blotting and immunofluorescence

Anti-mouse LAT4, as well as anti–mouse LAT4 phosphoS274 (pS274) and phosphoS297 (pS297) antibody production and specificity testing was described previously [11, 41]. For detection of the phosphorylated S6 ribosomal protein (S235/S236), a commercial rabbit mono-clonal antibody was used (Cat# 4858, Lot# 16, RRID:AB_916156, Cell Signaling Technology, Danvers, MA, USA). Total PKA phosphorylated (pS/pT) substrates were also detected using a rabbit polyclonal antibody (Cat# 9621, Lot# 14, RRID:AB_330304, Cell Signaling Technology, Danvers, MA, USA). To detect β-actin, a commercially available mouse monoclonal antibody was used (Cat# A2228, Lot# 067M4856V and #118M4829, RRID:AB_476697, Sigma-Aldrich, Buchs, Switzerland). For E-Cadherin detection, a commercially available rat monoclonal anti-body was used (Innovative Research Cat# 13–1900, Lot# UA2682691, RRID:AB_86571, Invi-trogen, Carlsbad, CA, USA).

As secondary antibodies for Western blotting (WB), either anti-rabbit IgG HRP conjugate (Cat# W4011, Lot# 0000340771, RRID:AB_430833) or anti-mouse IgG (H+L) AP conjugate (Cat# S3721, Lot# 0000312817, RRID:AB_430871) were used (both from Promega, Dübendorf, Switzerland). For immunofluorescence (IF), Alexa Fluor™ 594 donkey anti-rabbit IgG (Cat# ab96921, Lot# GR170389-3, RRID:AB_10680407, Abcam via Life Technologies, Carlsbad, CA, USA) and Alexa Fluor™ 488 goat anti-rat IgG (H+L) (Cat# A-11006, Lot# 1887148, RRID: AB_2534074, Thermo Fisher Scientific via Invitrogen, Carlsbad, CA, USA) polyclonal antibod-ies were used. Cell nuclei were detected with 4,6-diamidino-2-phenylindole (DAPI) (Cat# D3571, Lot# 778144, RRID:AB_2307445, Thermo Fisher Scientific via Invitrogen, Carlsbad, CA, USA).

## Immunofluorescence

Harvested small intestinal segments (duodenum, jejunum and ileum) from diurnal and dietary regulation experiments were cut open and flattened, then fixed for 1 hour in 4% PFA/PBS at 4˚C. Intestines were then "Swiss-rolled" [42] and fixed overnight in 4% PFA/PBS at 4˚C. "Swiss rolling" is a technique that allows to inspect and analyze long, uninterrupted pieces of small intestine (in our case 4–5 cm of each segment), giving more detailed information on changes between various intestinal segments and increasing the n of villi and crypts that can be analyzed, when compared to embedded intestinal rings or short pieces. After fixation, Swiss rolls were washed with PBS for 3 hours, incubated with 15% sucrose for 5 hours, followed by an overnight incubation in 30% sucrose, all at 4˚C. Afterwards, the intestinal rolls were placed into 30% sucrose-OCT embedding medium (Cat# 81-0771-00, Lot# 0381141, Biosystems, Muttenz, Switzerland) mix (1:1 ratio) for 1 h at room temperature before embedding in OCT on dry ice.

Consecutive 3–5 μm sections were prepared from the Swiss rolls and fixed on Superfrost®-Plus Menzel slides (Cat# J1800AMNZ, Lot# 030118, Gerhard Menzel GmbH, Braunschweig, Germany). Afterwards the sections were washed for 5 min: twice with PBS and once with hypertonic PBS (+18 g L⁻¹ NaCl). No epitope retrieval was performed for antibodies against

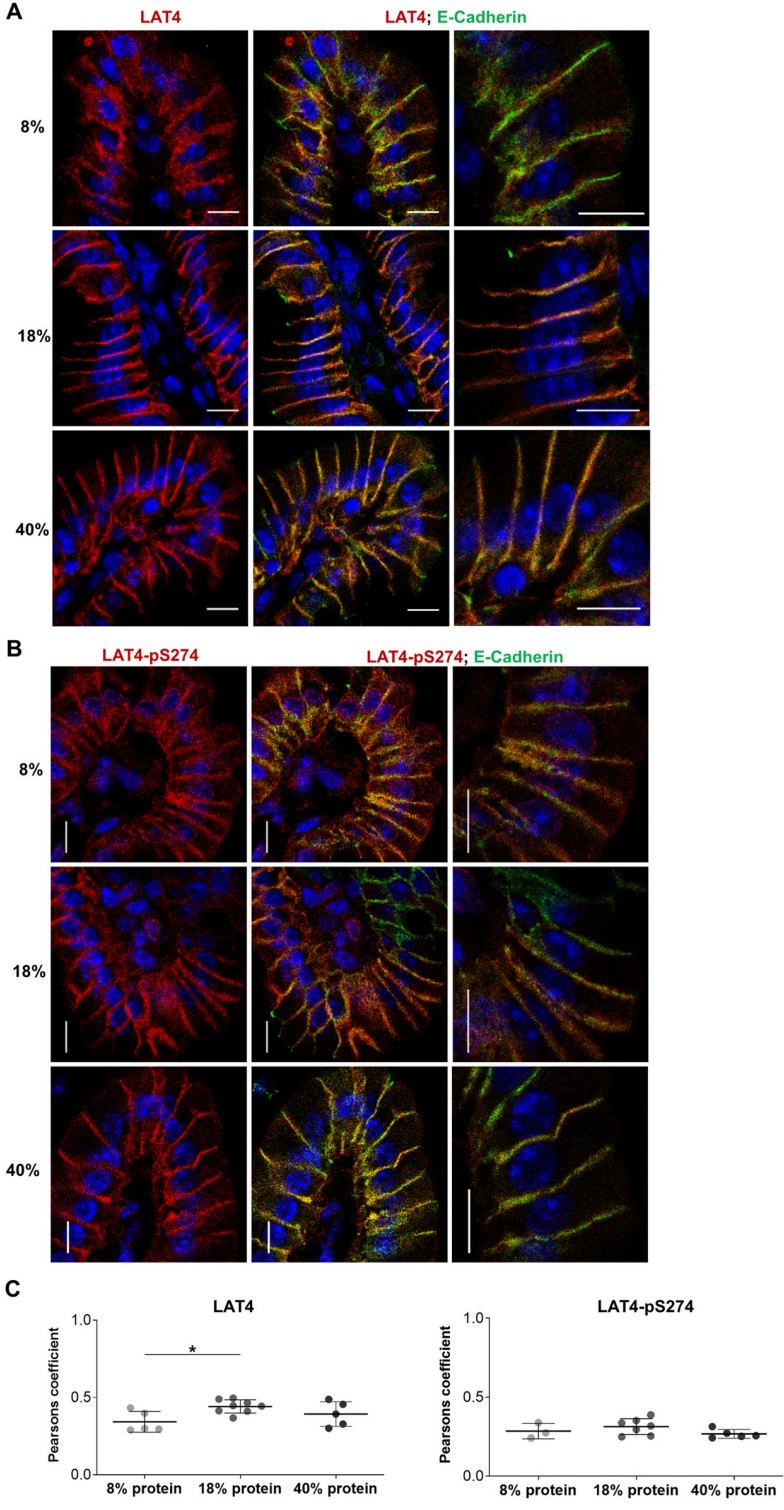

**Fig 9. Entrainment with low protein diet leads to increased intracellular localization of LAT4 in jejunum during food anticipation.** Villi were stained with antibodies against LAT4 (red) or pS274 (red), and membrane marker E-cadherin (green). Nuclei stained with DAPI (blue). **A**: Subcellular localization of LAT4. **B**: Subcellular localization of pS274. Scale bar: 8 μm. **C**: Pearson's colocalization coefficient was determined with ImageJ 1x EzColocalization plugin. Mean (SD) shown, n = 3–8 mice per diet from two separate experiments. 0 = no colocalization, 1 = complete colocalization. Statistical analysis: One-way ANOVA with Tukey's multiple comparison test, *p<0.05.

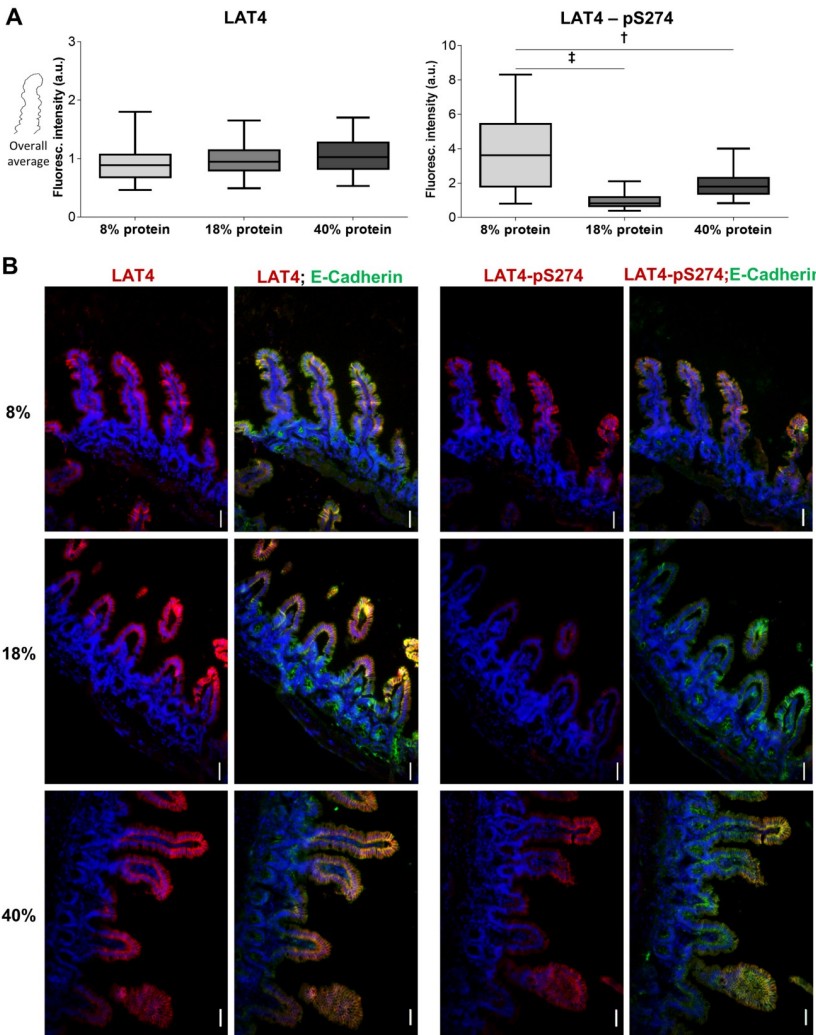

**Fig 10. Entrainment with low protein diet leads to increased S274 phosphorylation during food anticipation also in ileum. A**: Quantitation of fluorescence intensity of ileal villi, of mice entrained and euthanized at ZT12 as for Fig 8, stained with antibodies against LAT4 (left panel) or pS274 (right panel), and membrane marker E-cadherin. Fluorescence intensity determined with ImageJ 1x software and related to intensity of E-cadherin. Median shown, n = 170–320 villi from 4–5 mice per diet from two separate experiments. Statistical analysis: One-way ANOVA with Tukey's multiple comparison test, [†]p<0.01; [‡]p<0.001. **B**: LAT4 expression and phosphorylation on S274 in mice ileum. Red: LAT4 or pS274, green: E-Cadherin, blue: DAPI. Scale bar: 30 μm.

LAT4-pS274. Epitope retrieval for antibodies against LAT4 and pS6 was performed in sodium citrate, pH 6.0, for 10 min at 98°C using a microwave histoprocessor (HistosPRO, Milestone, Sorisole, Italy). After epitope retrieval, sections were washed again in PBS and incubated in blocking solution (2% bovine serum albumin, 0.04% Triton X-100 in PBS, pH 7.4) for 1 hour at room temperature (RT). Primary antibodies were also diluted in blocking solution and incubated overnight at 4°C. Dilutions used: anti-LAT4 1:1500, anti-pS274 1:1000, anti-pS6 1:500 and anti-E-Cadherin 1:300. Then the sections were washed as before and incubated with secondary antibodies diluted 1:2000 in blocking solution for 1 hour at RT in the dark. Together with the secondary antibodies, nuclei staining (DAPI 1:5000) was performed. Afterwards, sections were washed again and fixed using Glycergel Mounting medium (Cat# C0563, Lot#

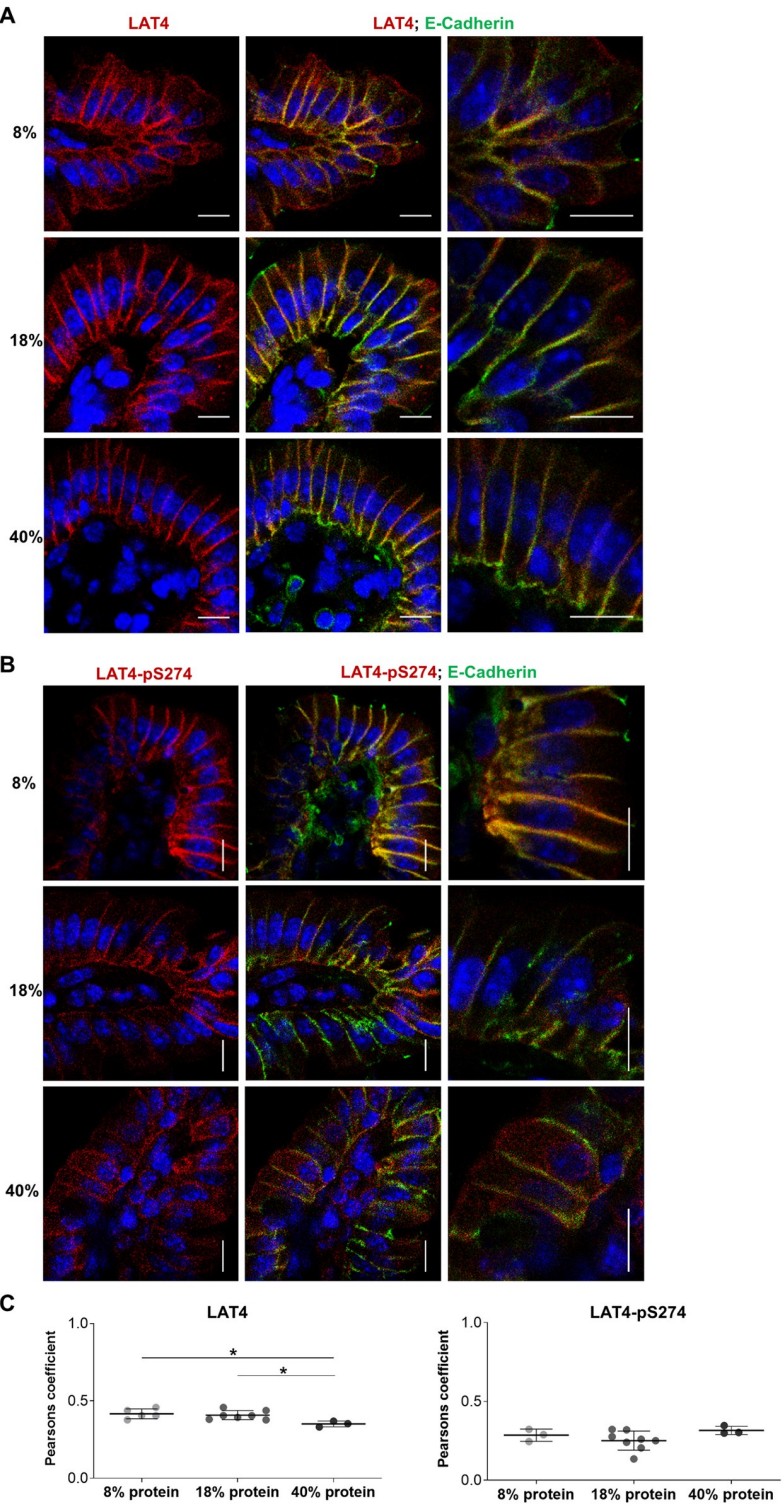

**Fig 11. Entrainment with high protein diet increases LAT4 intracellular localization in ileum during food anticipation.** Villi were stained with antibodies against LAT4 (red) or pS274 (red), and membrane marker E-cadherin (green). Nuclei stained with DAPI (blue). **A**: Subcellular localization of LAT4. **B**: Subcellular localization of pS274. Scale bar: 8 μm. **C**: Pearson's colocalization coefficient was determined with ImageJ 1x EzColocalization plugin. Mean (SD) shown, n = 3–8 mice per diet from two separate experiments. 0 = no colocalization, 1 = complete colocalization. Statistical analysis: One-way ANOVA with Tukey's multiple comparison test, $^*$p$<$0.05.

10139831, Agilent Technologies, Santa Clara, CA, USA). All samples were imaged using fluorescent light microscopes (Nikon Eclipse TE 300 and Leica SP5 confocal).

## Image analysis

To assess phosphorylation differences along the intestinal villus, their images were analyzed as described before [11]. Shortly, jejunal villi were split into 3 identical parts: tip, middle and base (Fig 4, upper right corner image—the white line marks the villus, the blue line indicates the applied border between the tip and the middle of the villus and the green line the border between the middle and the base), and the average fluorescence intensity of each part was measured for the different color channels. Ileal villi were measured in full. Values given by LAT4 or phosphorylation specific antibodies (pS274, pS6) were then related to the fluorescence intensity of the membrane marker E-Cadherin. For pS274 antibody, values were also normalized to total LAT4. The exposure time, gain, brightness and contrast were kept constant for all tissue sections stained with the same antibody. Only the full-length villi that were not damaged or segmented by section cutting and processing were used for the analysis, totaling to n = 25–60 villi per intestinal segment (lower values for ileum due to shorter segment length) per mouse.

To assess the colocalization between LAT4 or pS274 and the membrane marker E-cadherin, we imaged the full thickness of jejunal and intestinal villi cryosections as a z-stack at 63x magnification using a Leica SP5 confocal microscope. Afterwards a Z projection was made from the sum of 5 slices (1 slice = 0.5 μm) to cover the vertical mid-section of the villus. Red and green channels were separated and the colocalization was assessed by EzColocalization plugin [43], with Renyi's entropy used as threshold algorithm and by calculating Pearson's colocalization coefficients. All image analysis was done using ImageJ 1x software (ImageJ) (National Center for Microscopy and Imaging Research: ImageJ Mosaic Plug-ins, RRID:SCR_001935) [44].

## Western blotting

Total membranes from diurnal rhythm and ghrelin experiments or total lysate prepared from the isolated villi fractions (epithelial cell sheets) used for *ex vivo* treatments, prepared as described in the corresponding section above, were used for Western blotting. Protein concentration in the lysates was determined using Pierce™ BCA Protein Assay (Cat# 23228, Lot# UA269551, ThermoFischer Scientific, Rockford, IL, USA) according to the manufacturer's instructions. Afterwards 15 μg of protein were diluted in 4 x Laemmli buffer containing 10% β-Mercaptoethanol, incubated at RT for 20 min and loaded on 1 mm 10% polyacrylamide gel. Separation was done by electrophoresis and afterwards PVDF membrane (Cat# IPVH00010, multiple lots used, Immobilion-P, Merck Millipore, Burlington, MA, USA) was used for wet transfer. Membranes were blocked for 1 hour at RT with either 5% milk powder in Tris-Buffered saline containing 0.1% Tween-20 (TBS-Tween) (Tween-20, Cat# P1379, Lot# SLCB2671, Sigma-Aldrich, Buchs, Switzerland) or with 2% SureBlock (for phosphospecific antibodies) (Cat# SB232010, Lot#180705, LuBioscience, Lucerne, Switzerland) in TBS-Tween. Primary antibodies were diluted in 5% milk powder-TBS-Tween (anti-LAT4 1:2000) or in 2% Sure-Block-TBS-Tween (anti-pS274 1:750, anti-pS297 1:300, anti-pS/T PKA substrates 1:750) and incubated overnight at 4˚C. Afterwards blots were washed in TBS-Tween for 3 x 5 min and secondary antibodies applied in the same solution in 1:5000 dilution and incubated for 1 h at RT. β-actin in 1:5000 dilution was used as a loading control.

Antibody binding was detected using Luminata Classico Western Chemiluminescent HRP substrate (Cat# WBLUC0100, multiple lots used, Merck Millipore, Burlington, MA, USA) or

CDP-Star® substrate (Cat# NIF1229, multiple lots used, GE-Healthcare UK, Buckingham-shire, UK). Blots were imaged with ImageQuant® (ImageQuant, RRID:SCR_014246) LAS 4000 camera (Fujifilm, Tokyo, Japan). Densitometric analysis was performed on the original blots with ImageJ software. Exposure times used: 30 s for LAT4, pS274 and control β-actin antibodies, 1 minute for pS297 and PKA substrate antibodies. Signal intensity from the antibody of interest was normalized to the loading control β-actin. For pS274 and pS297 antibodies, this ratio was further normalized to the LAT4/ β-actin ratio. For visualization purposes, the brightness and contrast of the whole images shown in Figs 2–3, 12 and 13 and S1 Fig was adjusted using ImageJ.

## RNA extraction and real-time PCR

Freshly harvested mouse intestine was placed on a dry, ice-cold surface, and 1–2 cm from the duodenum was cut off, everted and scraped with a surgical scalpel to isolate the mucosa. Scraped mucosa was immediately frozen in liquid nitrogen. Total RNA was extracted using the RNeasy Mini Kit (Cat# 74106, Lot# 160031326, Qiagen, Hombrechtikon, Switzerland) according to the manufacturer's instructions, with the exception that samples were lysed in 1 ml of TRIzol™ Reagent (Cat# 15596026, multiple lots used, Ambion Life Technologies, Carls-bad, CA, USA) whilst still frozen. MagNa Lyser Green Beads (Cat# 03358941001, multiple lots used, Roche, Basel, Switzerland) were added to the samples and lysis was done at 6000 rpm for 2 x 30s in a Precellys 24 homogenizer (Cat# 03119.200.RD000, Bertin Technologies SAS, Montigny-le-Bretonneux, France). The quality and quantity of the extracted RNA were assessed with the RNA 600 Nano kit (Cat# 5067–1511, Lot# KF04BK03, Agilent Technologies, Santa Clara, CA, USA) and a Nanodrop ND-1000 UV spectrophotometer (NanoDrop Technologies, Wilmington, USA), respectively. Reverse transcription was performed with the qScript cDNA synthesis kit (Cat# 95047, Lot# 020927, Quantabio Inc., Beverly, MA, USA) using 500 ng of RNA as template. For real-time (RT) PCR 75 ng cDNA, TaqMan® Universal PCR Mastermix (Cat# 4304437, multiple lots used, Applied Biosystems, Waltham, MA, USA) and other reaction components were mixed in accordance to the manufacturer's instructions. Primers and probe for LAT4 were described previously [41]. Other primers and probes used are listed in Table 2. All were ordered from Microsynth (Balgach, Switzerland), with exception that the LAT1 probe was ordered from Roche Universal Probe library (#34, Roche, Basel, Switzerland). The abundance of the target mRNA (test RNA) was normalized to the 18S ribosomal RNA (18S RNA). All reactions were performed in triplicates and relative expression ratios were calculated as $R = 2^{-(Ct(test\ RNA)-Ct(18S\ RNA))}$, where Ct is the cycle number at the threshold for the tested mRNAs.

## Statistics

Experimental data analysis was performed using GraphPad Prism v8.2 (Graphpad Prism, RRID:SCR_002798, GraphPad software, San Diego, CA, USA). Differences between the mean values of two groups were assessed with unpaired Student's t-test (Fig 1B). For assessment between more than two groups One-way analysis of variance (ANOVA) was done, followed by Tukey's multiple comparison test (Figs 2–3, 5 and 13 and S1 Fig). Individual mRNA expression levels (Fig 1B), all Western blot quantifications (Figs 2–3, 12 and 13 and S1 Fig) and calculated Pearson's coefficient values (Figs 9C and 11C) are shown as separate points representing each mouse with mean group value and standard deviation (SD) indicated with lines.

One-way ANOVA for the quantified immunofluorescence (LAT4, pS274 and pS6) of intestinal villi (for pS6 also crypts) (Figs 5, 6A, 7A–7F, 8 and 10) was done considering measurements from each individual mouse. Data were analyzed as a "Grouped" data set on GraphPad

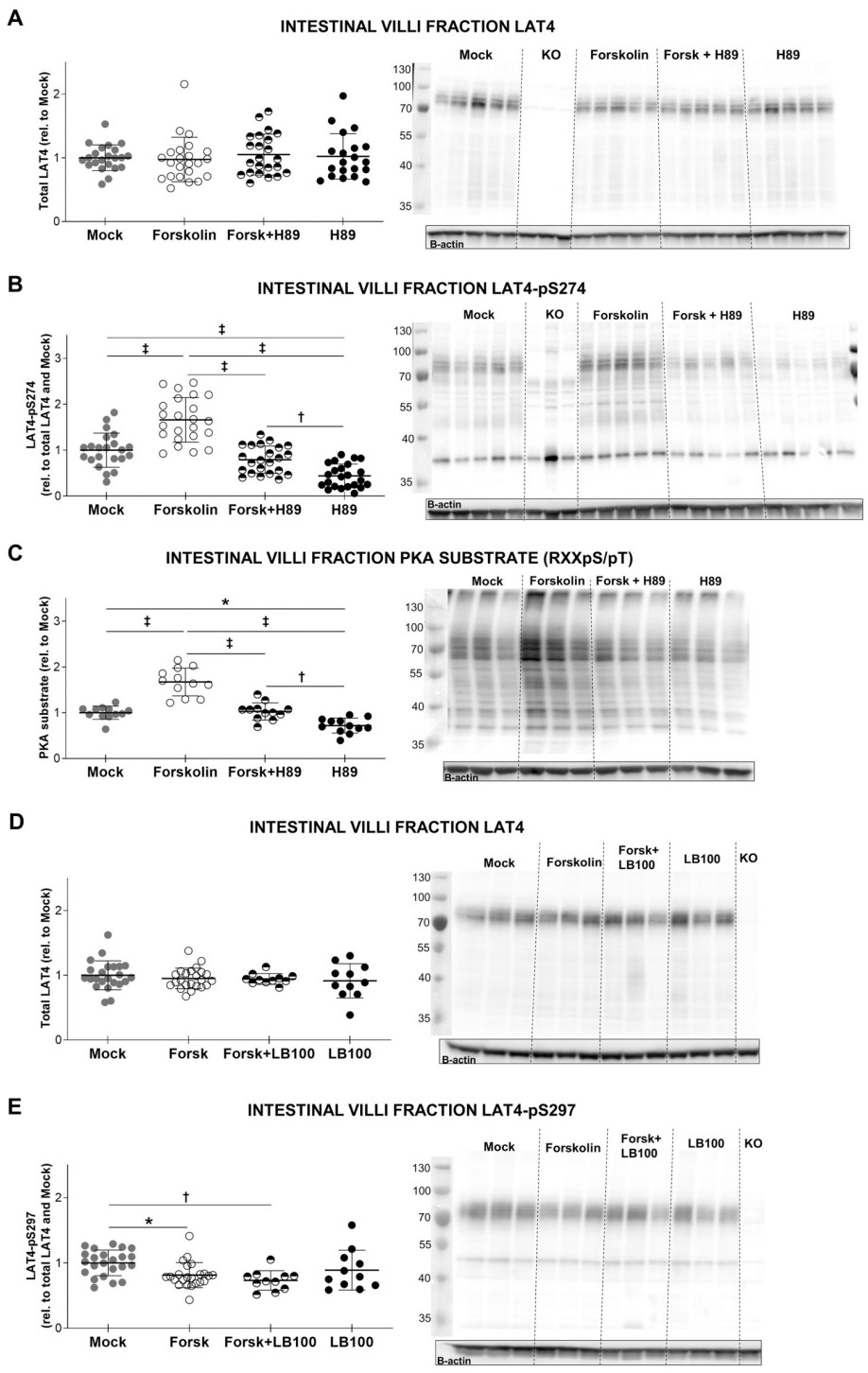

**Fig 12. Protein kinase A presumably downregulates LAT4 function by increasing its phosphorylation on S274 and decreasing pS297 level.** Intestinal villi fraction was isolated from the whole small intestine of overnight starved mice and treated *ex vivo* for 10 minutes with forskolin to activate PKA and/or H89 (PKA inhibitor) and/or LB100 (PP2A inhibitor). **A**: LAT4 protein expression. **B**: LAT4 phosphorylation on S274. **C**: Phosphorylation level of PKA substrates. **D**: LAT4 protein expression. **E**: LAT4 phosphorylation on S297. **A to E**: Total lysates from intestinal villi fraction of WT and LAT4 conditional KO mice analyzed by Western blot. For quantification, all values were normalized to beta actin. Further normalization to LAT4 ratio was done for phospho-specific antibodies against pS274 and pS297. Quantification is shown in the left panel, representative blot—in the right panel. Statistical analysis performed with One-way ANOVA using Tukey's multiple comparison test, $^*$p<0.05; $^†$p<0.01, $^‡$p<0.001. Mean (SD), n = 22–23 mice

per treatment from four separate experiments (LAT4, pS274, pS297); n = 11–12 (PKA substrates) and n = 10–11 (LB100 treatments) mice per treatment from two separate experiments.

with 5–6 mice per time point or 4–5 per specific diet, entering all individual villus/crypt fluorescence values measured for each mouse (n = 25–60). This was done to include observed high individual variances into the analysis and to avoid skewing towards higher statistical differences as would arise if whole time point or diet group values would be analyzed together as a "Column" data set. However, for simpler visualization, in figures all individual values were pooled and shown as a box (25th and 75th percentile) and whiskers (5th to 95th percentile) plot with a line showing the median.

To detect and visualize diurnal oscillations, cosinor analysis was performed for all Western blot quantifications in duodenum and jejunum (LAT4, pS274, pS297—all measurement values entered) and for the total immunofluorescence quantifications in ileum (LAT4, pS274, pS6—only average value per each mouse per time point was entered due to high number of values), using Cosinor.Online program [45] (available at https://cosinor.online/app/cosinor.php). Regarding pS6 data from ileum, cosine fit was done excluding the ZT8 time point and shown for villi and crypts separately (Fig 7E and 7F). The generated cosine waves were superimposed on the actual measurement data (Figs 2, 3, 6 and 7) and their fit determined by zero-amplitude test.

## Results

### Small intestinal amino acid uniporter LAT4 does not undergo food-entrained diurnal mRNA regulation in mouse unlike symporters and antiporters

We have previously shown that the mRNA of amino acid uniporter LAT4 in food-entrained mouse jejunum exhibits no significant diurnal difference between resting phase and the start of the active phase, thus showing no food anticipatory response at this level [11]. Based on recent studies demonstrating high mRNA expression levels of multiple solute carriers in the duodenum [6, 7], we tested whether mRNA of LAT4 and of other nutrient transporters could be diurnally regulated in this segment. We food-entrained WT mice (CreERT2$^-$ with floxed *Lat4*) by feeding them for only 8 hours during the dark, active phase (ZT12 to ZT20) over a period of at least two weeks with a standard 18% protein diet (Fig 1A). Mice were then euthanized either at the beginning of the resting (ZT0) or of the active (ZT12) phase and scraped duodenal mucosa was collected to test the mRNA expression level of different nutrient transporters. It is important to note that at ZT12, mice were euthanized before the start of feeding, in order to detect possible food anticipation.

In this setting, we observed that amino acid symporters B$^0$AT1 (Slc6a19) and glucose symporter SGLT1 (Slc5a1) show a clearly increased mRNA expression at the start of the active phase (ZT12). This diurnal regulation of nutrient symporters has been known from earlier reports [46–48] and thus demonstrated successful food entrainment and synchronization of intestinal clocks (Fig 1B, right panel). Similarly, the mRNAs of all tested nutrient antiporters and possibly also that of the membrane glycoprotein 4F2hc (Slc3a2) showed the same anticipatory upregulation (Fig 1B, bottom panel). In contrast, all tested uniporters, including LAT4, exhibited no clear anticipatory mRNA upregulation, with a possible exception for glucose transporter GLUT2 (Slc2a2) (Fig 1B, left panel). This suggests that under food entrainment, active and passive nutrient transporters, especially the amino acid uniporters (LAT4, LAT3, TAT1), may be submitted to different regulatory mechanisms.

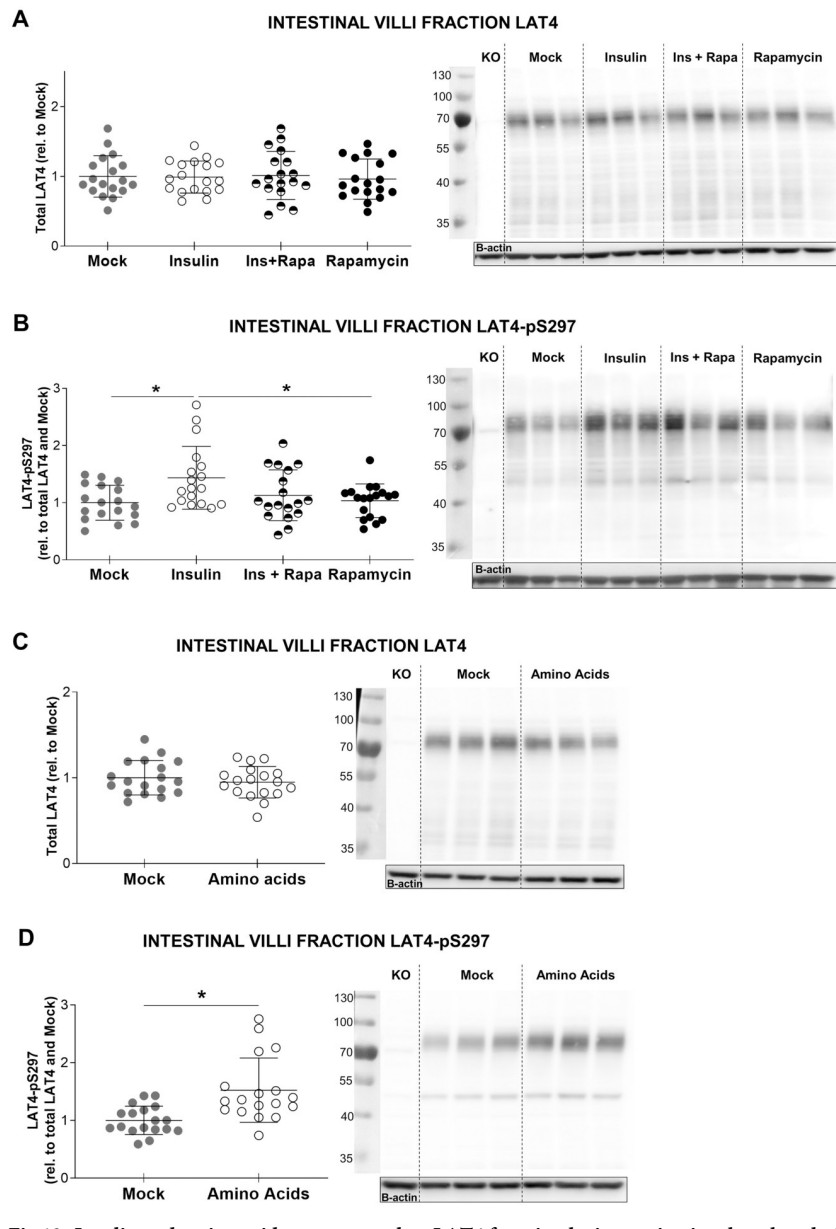

**Fig 13. Insulin and amino acids may upregulate LAT4 function by increasing its phosphorylation on S297.**
Intestinal villi fraction was isolated from the whole small intestine of overnight starved mice and treated *ex vivo* for 10 minutes with insulin and/or rapamycin, or an amino acid mix (5-fold concentrated relative to plasma). **A**: LAT4 protein expression under insulin treatment. **B**: LAT4 phosphorylation on S297 under insulin treatment. **C**: LAT4 protein expression under amino acid treatment. **D**: LAT4 phosphorylation on S297 under amino acid treatment. **A to D**: Total lysates from intestinal villi fraction of WT and LAT4 conditional KO mice analyzed by Western blot. For quantification, all values were normalized to beta actin. Further normalization to LAT4 ratio was done for phospho-specific antibodies. Quantification is shown in the left panel, representative blot in the right panel. Statistical analysis performed with One-way ANOVA using Tukey's multiple comparison test, *p<0.05; †p<0.01. Mean (SD), n = 17–18 mice per treatment from three separate experiments.

**Table 2. Primers and probes used for nutrient transporter mRNA detection.**

| mRNA | Accession Nr. | Forward primer | Reverse primer | Probe |
|---|---|---|---|---|
| Lat1 (Slc7a5) | AB017189.1 | TTTGCTTGGCTTCATCCAGAT | GGACAACTTCTGCTGCAGGTT | AAGGACATGGGACAAGGTGATGCGTC |
| Lat2 (Slc7a8) | NM 016972 | TCCACGTTTGGTGGAGTCAAT | TGGATCATGGCCAACACACT | CTCCCTCTTCACCTCCTCCCGGCT |
| Lat3 (Slc43a1) | NM_001081349 | CCCTGAATGAGAATGCTTCCTT | ATGGCATTGGTGAGCTTTTGT | AGCACCAAGTTCACTAGACCACGCTACCG |
| y$^+$Lat1 (Slc7a7) | NM 011405 | AATTCCAGTAGCGGTTGCATT | GGAGGTGGCCTTCTCTCGAG | TTGCTTTGGTGGGCTCAACGCC |
| Tat1 (Slc16a10) | NM_028247.4 | CGCCTACGGGGTGCTCTTC | ACTCACGATGGGGCAGCAG | CGAGCCCACCCACGCTGTCTTG |
| 4F2hc (Slc3a2) | NM_008577 | GTTTTTGAATGCCACTGGCA | GTCCTGAGGAGCGTCTGAAA | ATGGTGCAGCTGGAGTGTGTCGCA |
| B$^0$at1 (Slc6a19) | NM_028878.3 | GCCACTGAGCGCTTTGATG | GCCTCAAAGTTCTCTGAAGTCACA | ATGGGTTCGACCTGCCGGAGG |
| Sglt1 (Slc5a1) | NM_019810 | GTTGGAGTCTACGCAACAGCAA | GGGCTTCTGTGTCTATTTCAATTGT | TCCTCCTCTCCTGCATCCAGGTCG |
| Glut2 (Slc2a2) | NM_031197.2 | GGCCCTTGTCACAGGCATT | CCTGATTGCCCAGAATAAAGCT | TTATTAGTCAGATTGCTGGCC |

## Protein expression and phosphorylation of amino acid uniporter LAT4 are submitted to food-entrained diurnal regulation in mouse duodenum

WT mice from C57B6/J background were food-entrained as shown in Fig 1A and euthanized every 4 hours with subsequent isolation of the intestinal villi fractions. Diurnal regulation of LAT4 protein expression and phosphorylation on S274 (pS274) and S297 (pS297) was tested by Western blotting. Villi fractions from induced LAT4$^{flx/flx}$ROSACreERT2$^+$ knockout mice were used as negative controls for the phosphospecific antibodies.

Whereas we did not detect any difference in duodenal LAT4 mRNA expression between resting and active period (Fig 1B), we observed that the protein expression of LAT4 tends to increase towards and during the active period, reaching significantly higher levels in the middle of the feeding phase (ZT16) (Fig 2A). A cosine wave fit with period of 24 hours indicated that the acrophase of LAT4 expression occurred between ZT13 and ZT14 –during the first hours of food intake. The fit of the wave was confirmed using the zero-amplitude test (p = 0.011) (Table 3).

The observed discrepancy between mRNA and protein expression might indicate that the expression of LAT4 protein in response to food entrainment could be regulated at the level of the translation or post-translationally (e.g. protein stability).

In duodenum, phosphorylation on S274 also showed a food-entrained diurnal rhythmicity reaching lowest levels during food anticipation (ZT12) and intake (ZT16) (Fig 2B). Cosine wave fit placed the bathyphase between ZT13 and ZT14—during the first hours of food intake,

**Table 3. Cosine wave fit (period = 24 h) to characterize the diurnal rhythms.**

| Part of intestine | Fitted | Mesor | Acrophase (ZT h) | Bathyphase (ZT h) | Zero-amplitude test | |
|---|---|---|---|---|---|---|
| | | | | | F-value | P-value |
| Duodenum | LAT4 | 1.36 | 13.8 | 1.8 | 4.8 | 0.011 |
| | pS274 | 0.69 | 1.7 | 13.7 | 7.4 | 0.001 |
| | pS297 | 1.18 | | | 2.7 | 0.077 |
| Jejunum | LAT4 | 1.02 | | | 1.4 | 0.254 |
| | pS274 | 0.80 | 3.6 | 15.6 | 3.6 | 0.034 |
| | pS297 | 1.19 | 13.0 | 1.0 | 4.5 | 0.015 |
| Ileum | LAT4 | 0.96 | | | 1.2 | 0.311 |
| | pS274 | 0.63 | 1.8 | 13.8 | 37.0 | <0.00001 |
| | pS6 (Villi) | 0.37 | 18.7 | 6.7 | 14.7 | 0.00008 |
| | pS6 (crypts) | 0.65 | | | 3.3 | 0.05345 |

same as for increased LAT4 expression; the fit was confirmed by zero-amplitude test (p = 0.001) (Table 3). Based on our previous observations with *Xenopus* oocytes as an expression system, decreased S274 phosphorylation leads to increased affinity and thus to increased transport function of LAT4 [11]. We did not observe a significant coordinated diurnal regulation of S297 phosphorylation (p = 0.077) (Table 3). However, there was an increase in the individual variation of pS297 during the resting phase (ZT4 and ZT8) (Fig 2C). Interestingly, the highest and lowest values were not shown by the same mice at ZT4 and ZT8. Thus, it is not excluded that the high individual variability represents some regulatory process during which LAT4-S297 goes through phosphorylation and dephosphorylation cycles. We have shown previously, that phosphorylation on S297 might be necessary for LAT4 stability and function— dephosphorylation at this site led to a decrease in the affinity and the maximal transport rate of LAT4 (*Xenopus* expression system) and almost abolished its transport function [11]. We can however only assume that the functional impact of these phosphorylation changes is similar *in vivo* as previously observed in the *Xenopus* oocyte expression system. We are indeed not aware of any method or system that would allow us to directly test *in vivo* the functional impact of a specific phosphorylation change. As possible alternatives, we tested several *ex vivo/ in vitro* systems using cultured epithelial cells, intestinal organoids and isolated villi, but none of them was mimicking the *in vivo* situation satisfactorily.

## Only the phosphorylation of amino acid uniporter LAT4 is responsive to food entrainment in mouse jejunum and ileum

Using total membrane lysates of the isolated jejunal villi fractions (Western blotting) and Swiss-roll cryosections (Immunofluorescence) from the same food-entrained mice, we performed a detailed investigation of the diurnal regulation of LAT4 and its phosphorylation in jejunum. In ileum we prepared only Swiss rolls, due to its short length and short villi, and visualized the extent and pattern of S274 phosphorylation. S297 phosphorylation could not be investigated in this segment due to the lack of immunoreactivity of our antibody on cryosections [11].

We had shown previously in jejunum that LAT4 protein expression is not under food-entrained diurnal regulation (ZT0 vs ZT12), but that its phosphorylation on S274 was decreased during food anticipation (ZT12), while that of S297 tended to increase [11]. Now we additionally demonstrate that changes in LAT4 phosphorylation become even stronger in the middle of the feeding period (ZT16) (Fig 3). We also confirmed the lack of significant food-entrained diurnal rhythmicity in LAT4 expression in jejunum by poor cosine wave fit (zero-amplitude test, p = 0.254) (Table 3). Phosphorylation on S274 reached its lowest level during food intake, as indicated by the calculated bathyphase between ZT15 and ZT16 (cosine wave fit confirmed by zero-amplitude test, p = 0.034) (Fig 3B) (Table 3). Phosphorylation on S297 did show high individual variability, similar to our observations in duodenum (Fig 2C), however the cosine wave fit did pass the zero-amplitude test (p = 0.015) and indicated that its acrophase occurs between ZT13 and ZT14 (Table 3), thus during the first two hours of food intake. These data suggest that the phosphorylation of jejunal LAT4 not only shows changes due to entrained food intake anticipation, but potentially also responds to the actual luminal content during food intake.

We again observed the patchy localization of pS274 (Fig 4), previously seen at ZT12 in jejunum [11]. Current results showed that patchiness starts to increase already before the end of the resting period (~ZT8), reaching its peak at ZT12 and ZT16. Staining of pS274 showed the strongest decrease in the middle part of the villi or towards their base (Figs 4, 5A and 5B).

As described above, the overall decrease in S274 phosphorylation at the beginning of the feeding period was similarly detected by two different techniques, namely Western blotting and immunofluorescence. Since these techniques involve different sample preparations, labeling and quantifications, the validity of the results is strongly supported. However, since only certain parts of villi seem to lose the phosphorylation, the changes in total pS274 are not as severe (Figs 3B and 5C).

In ileum we also observed no significant food-entrained diurnal regulation of LAT4 protein expression, supported by a lack of difference between the cosine wave and the zero-amplitude wave (p = 0.311) (Table 3). However, ileum showed an even more drastic reduction of S274 phosphorylation than jejunum during food anticipation and intake (ZT8 –ZT20), with the bathyphase between ZT13 and ZT14 (cosine fit confirmed by zero-amplitude test, p<0.00001) (Fig 6A, right panel and Fig 6B) (Table 3). Indeed, in many cases pS274 labeling disappeared along the entire villus height during the feeding time. In the villi in which pS274 signal remained, it mostly localized towards the villus tip (Fig 6B).

Overall, in the jejunal and ileal villi of food-entrained mice, LAT4 shows location-specific changes in its phosphorylation, with possible food anticipation starting hours before the actual feeding period. The reproduction of previously observed food anticipatory response [11] with other, larger cohorts of mice strongly strengthens these results. In addition, during the feeding period these changes appear to be increased, suggesting the possibility that phosphorylation responses to anticipation and feeding might be mediated via different pathways.

## Mammalian target of rapamycin complex 1 (mTORC1) shows a strong response to food-entrainment in ileum, but not in jejunum

Phosphorylated ribosomal protein S6 (pS6) is an important downstream effector of mTORC1 and its phosphorylation is thus considered to be one of the major markers of mTORC1 pathway activity [18, 49]. In preliminary tests we had observed that the phosphorylation of S6 also localizes in patches along the intestinal villi. To see if there were any location similarities between the pS6 and pS274 patches, we stained Swiss-roll cryosections from the jejunum and ileum of food-entrained mice with pS6 antibody. However, in jejunal villi we did not observe any similarities in patchiness of pS6 and pS274 (Figs 4, 5 and 7A–7C). Interestingly, besides a slight decrease in total pS6 levels in both villus and crypts four hours after the start of the resting phase (ZT4) (Fig 7C and 7D), we did not observe any other clear difference at the level of jejunal pS6 between the time points.

In contrast, in the ileum we detected a strong food-entrained response of S6 phosphorylation (Fig 7E–7G). In villi during the resting phase we observed first a decrease of pS6 staining at ZT4 that was followed by a surprisingly high and transient phosphorylation signal at ZT8. When these high pS6 values observed at ZT8 were excluded, we were able to fit a cosine wave to the rest of the pS6 data, with the acrophase occurring between ZT18 and ZT19 –towards the end of the feeding phase (curve fit confirmed with zero-amplitude test, p = 0.00008), indicating underlying diurnal rhythmicity (Table 3) (Fig 7E).

Ileal crypts showed a similar pattern of phosphorylation changes, with the highest pS6 level during the food intake period (ZT16-ZT20) and with the strongest staining at the bottom of the crypts. Fitting a cosine wave to the pS6 values excluding the ZT8 time point yielded a curve that compared with a zero-amplitude wave displayed a p-value that was slightly above the significance limit (p = 0.053) (Table 3) (Fig 7F).

It has been shown that in mice small intestine, phosphorylation on S6 is low during the fasted state and increases upon feeding [50], similar to our observations for ZT4 and ZT16 and overall diurnal rhythmicity of pS6 in ileal villi. However, the transient increase in pS6 levels

observed in ileum during the resting phase at ZT8, 12 hours after food removal, has to our knowledge not been described before. It might be that this transient mTORC1 activation is induced by an increase in intracellular amino acid concentration mediated by an activation of autophagy and/or basolateral amino acid uptake from the blood. However, the origin of this transient mTORC1 activation and the reason why it was observed only in ileum remains unclear for us.

## Dietary protein content affects LAT4 phosphorylation and subcellular localization in jejunum and ileum

Following the investigation of LAT4 food-entrained diurnal phosphorylation changes, we addressed the question of LAT4 response to dietary protein content. WT mice (CreERT2⁻ with floxed *Lat4*) were food-entrained for two weeks (as in Fig 1A) with either 8% (low), 18% (standard) or 40% (high) protein diet and euthanized at ZT12 to compare changes in the entrained food-anticipatory response. In this experiment we used only immunofluorescence to assess LAT4 protein expression, localization and phosphorylation on S274. For technical reasons (only a short length of each intestinal part was available, the rest was collected for strain characterization) it had not been possible to prepare intestinal villi fractions from these mice such that only scraped intestinal mucosa was collected. Previously, we already had encountered issues with excessive proteolysis of LAT4 in scraped mucosa samples [11]. Unfortunately, also this time we found the quality of samples prepared for Western blotting unsatisfactory, due to high levels of LAT4 degradation. Since pS297 antibody shows a specific signal only when used for Western blotting [11], we were not able to assess changes in S297 phosphorylation in dietary protein intake experiments.

In jejunum, the fluorescence signal obtained with LAT4 antibody tended to be less abundant towards the base of the villi (expression ratio middle versus base >1 in Fig 8B left panel) and its quantitation revealed only minor differences in localization along the villi of mice food-entrained with different diets (Fig 8A, 8B and 8D left panels). However, the total LAT4 protein expression measured during food expectation was significantly increased in mice entrained with 40% protein diet relative to mice entrained with the standard 18% protein diet or the low 8% protein diet (Fig 8C left panel).

In contrast to total LAT4 protein, its phosphorylation in position S274 measured during food anticipation was increased in jejunum under low protein diet and showed a more uniform distribution along the villi (with some decrease towards the villus base) (Fig 8A–8D, right panels). Regarding the subcellular localization, we observed that entrainment with low protein diet led to poor co-localization of LAT4 protein with E-cadherin, as indicated by significantly lower Pearson's colocalization coefficient (PCC) (0.34 vs 0.44; p = 0.03) and an increase in intracellular signal from LAT4 antibody (Fig 9), compared to entrainment with normal protein diet. These results show that during food anticipation under entrainment with low protein diet, both LAT4 phosphorylation on S274 and its intracellular localization increase, possibly suggesting a downregulation of LAT4 function. In contrast, the entrainment with normal or high protein diet led to similar LAT4-pS274 levels and subcellular localization (Figs 8C, 8D and 9). However, in general LAT4-pS274 showed a poor co-localization with the membrane marker E-cadherin under all diets, suggesting that this site is mostly phosphorylated in the intracellular LAT4 fraction (Fig 9).

In ileum, LAT4 protein expression seemed not to be affected by the dietary protein content (Fig 10A and 10B left panels), however its colocalization with the membrane marker E-Cadherin and thus its surface localization appeared to decrease slightly under entrainment with high protein diet, when compared with both low and normal protein diets (PCC 0.35 vs 0.41

and 0.42, p = 0.03/0.02, respectively) (Fig 11A and 11C). An increase of LAT4-pS274 level was observed in mice entrained with low protein diet (Fig 10A and 10B right panels), suggesting a similar decrease in LAT4 function as suggested above for jejunum. Interestingly, there was also a trend for increased S274 phosphorylation under entrainment with high protein diet when compared to normal protein diet (Fig 10A and 10B, right panels).

It is interesting that increasing the dietary protein content above the level of the standard 18% chow appeared to produce little further changes in LAT4 phosphorylation (jejunum) or expression (ileum) during food anticipation. This suggests that abundance, surface localization and function of LAT4 are upregulated by its substrate(s) only up to a threshold that is already reached with the standard 18% protein diet.

## Protein kinase A increases phosphorylation on LAT4-S274

Since our study focuses on S274 (de-)phosphorylation in response to both diurnal and dietary stimuli, we investigated the possible involvement of specific signaling pathways that might directly lead to the (de-)phosphorylation of this serine residue. PhosphoMotif Finder, an online tool from the Human Protein Reference Database [37] and PhosphoNet, a human phosphosite KnowledgeBase developed by Kinexus, indicated that S274 matches the protein kinase A (PKA) phosphorylation motif. Thus, we designed an *ex vivo* approach to test this possibility. We prepared intestinal villi fractions from WT (B6/J background) mice previously starved overnight. Due to the number of villi needed for the simultaneous treatments and controls, we were unfortunately not able to separate duodenum, jejunum and ileum, and had to use the whole small intestine. We performed first series of *ex vivo* treatments on these villi fractions using forskolin (25 μM) to activate PKA and/or H89 (30 μM) to inhibit it. Treatment with vehicle only was used as control (mock). Due to the progressive LAT4 degradation in *ex vivo* conditions, we were able to perform only short treatments and pre-treatments with inhibitors.

Forskolin and H89 treatment had no effect on LAT4 protein levels (Fig 12A), but both affected its phosphorylation on S274 (Fig 12B). Under forskolin treatment, S274 phosphorylation was increased and in the presence of H89 it was unchanged or slightly decreased compared to mock treatment. We observed similar phosphorylation changes for various PKA substrates (Fig 12C). In contrast, the S297 phosphorylation site showed minor dephosphorylation in the presence of forskolin (Fig 12D and 12E). Knowing that protein phosphatase 2A (PP2A) has been shown to be activated by PKA [51], we used the PP2A-specific inhibitor LB100 (10 μM) [52] to test its possible involvement in S297 dephosphorylation. However, pS297 levels did not increase in the presence of the PP2A inhibitor (Fig 12D and 12E). In conclusion, these results showed that a) PKA activation or inhibition was successful in our *ex vivo* villi system; b) LAT4-S274 might be a direct target of PKA and c) PKA might induce dephosphorylation of S297 via activation of a protein phosphatase which is presumably not PP2A. Both S274 phosphorylation and S297 dephosphorylation would, according to our previous oocyte experiments [11], indicate that active PKA mediates a decrease in LAT4 function.

## Insulin and amino acids increase phosphorylation on LAT4-S297

We tested whether LAT4 could be regulated by the gastrointestinal hormones ghrelin (involved in food anticipation) and insulin (response to food intake), and by its own substrates —amino acids.

Ghrelin concentration in plasma has been shown to exhibit a strong circadian rhythmicity in mice and to induce a food anticipatory behavior [30, 31]. Since ghrelin treatments can be done *in vivo*, we tested whether a ghrelin injection during the rest phase at ZT0 would induce a LAT4 phosphorylation resembling that observed during food anticipation at ZT12. Ghrelin

(10 μg) or saline were injected i.p. at ZT0 and ZT12 and animals sacrificed 15 minutes later. Activity of ghrelin was verified by measuring the expected increase in plasma growth hormone concentration (S1A Fig). Intestinal villi fractions were then prepared and LAT4 expression and its phosphorylation levels tested by Western blotting. This experiment revealed no ghrelin-induced changes (S1B–S1D Fig), such that ghrelin appears not to be involved in the food anticipatory LAT4 phosphorylation response.

We then tested a possible effect of insulin and amino acids on LAT4, submitting villi fractions to an *ex vivo* treatment with 10 nM insulin or an amino acid mix containing 5 x the physiological plasma concentration. These treatments affected neither LAT4 overall amount (Fig 13A and 13C) nor the level of its S274 phosphorylation. In contrast, both treatments induced a slight increase in LAT4 phosphorylation in position S297 (Fig 13B and 13D). Additionally, we observed that the insulin-induced phosphorylation on S297 shows some sensitivity to rapamycin (100 nM) (Fig 13B). These results suggest that in the small intestine, LAT4-S297 could be a downstream target of insulin-activated mTORC1, and activation of this signaling pathway by food intake may increase LAT4 function and thus amino acid absorption.

## Discussion and conclusions

In this study, we describe the food-entrained diurnal and dietary control of amino acid uniporter LAT4 phosphorylation on serine 274 and 297 in mice and suggest the involvement of PKA and insulin/mTORC1 signaling pathways based on *ex vivo* experiments.

### Regulatory differences between uniporters and active transporters

The mRNA levels of luminal active nutrient transporters—glucose/Na$^+$ symporter SGLT1 and amino acid/Na$^+$ symporter B$^0$AT1 -, have been reported to be increased during the active phase in rodent intestine [46, 47, 53]. In the current study, we observed the same regulatory pattern for both symporters, confirming successful food entrainment of our experimental mice (Fig 1B). Importantly, all other tested active transporters, namely symporters and antiporters, including the basolateral ones and the accessory protein 4F2hc, showed a similar regulation at the mRNA level (higher level at ZT12 compared to ZT0).

In contrast, the amino acid uniporters LAT4 and TAT1 (transport via facilitated diffusion) did not show such differences between ZT12 and ZT0 in food-entrained mice. The only tested uniporter that showed a trend for higher mRNA at ZT12 was the glucose transporter GLUT2 (Fig 1B). These results suggest differences in diurnal regulation of passive (uniporters) and active (symporters and antiporters) amino acid transporters under food entrainment. We propose that food-entrained diurnal regulation of amino acid uniporters in the small intestine might be mostly posttranslational, involving trafficking of synthesized transport proteins between submembranous stores and the plasma membrane. This hypothesis is based on our current and previous findings regarding the regulation of basolateral amino acid uniporter LAT4 [11] and is supported by its similarity with the insulin-sensitive glucose uniporter GLUT4 (Slc2a4). GLUT4 is expressed in skeletal and cardiac muscle and adipose tissue [54] and has striking structural and functional similarities with LAT4. Both LAT4 and GLUT4 are glycoproteins with 12 transmembrane domains (TMD), and a large intracellular loop between TMD 6 and 7 containing numerous phosphorylation (including S274 and S297 for LAT4 and S274 for GLUT4), ubiquitination and other posttranslational regulation sites [55, 56]. Both transporters have similar low affinity towards their substrates with $K_m$ around 5 mM [41, 55, 57, 58] and appear to have a long half-life [59] (A. Rajendran, unpublished observation).

GLUT4 insulin-regulated trafficking between cytoplasmic vesicles and the plasma membrane has been studied in much detail [55, 60, 61]. As regards LAT4, we have previously

shown in *Xenopus* oocytes that mimicking S274 phosphorylation leads to an increased intracellular localization, whereas the dephosphorylation-mimicking mutant was localized mostly at the plasma membrane [11]. In the current study, an increased LAT4 intracellular localization was observed during food anticipation in mice entrained with low versus normal protein diet in jejunum and correlated with a high pS274 level (Fig 8C right panel, Fig 9A and 9C left panel). Interestingly, also GLUT4 trafficking was shown to be regulated by phosphorylation on the same site—in *Xenopus* oocytes insulin activated kinase SGK1 phosphorylated GLUT4 on its S274, increasing its trafficking to the plasma membrane and glucose transport [62]. Even though the effect of S274 phosphorylation in LAT4 and GLUT4 appear to be opposite, in both cases changes in this serine phosphorylation seem to be involved in protein trafficking. Regarding the diurnal regulation in mice, muscle GLUT4 protein level has been shown to stay constant throughout the active and resting phases, similar to jejunal and ileal LAT4 (Figs 3A and 6A) [63]. Overall, our observations with LAT4 in the small intestine under food entrainment suggest a similar regulatory mechanism as for GLUT4 with possible storage in submembranous vesicles and substrate-sensitive trafficking to the plasma membrane. However, it remains speculative whether other amino acid uniporters, in particular TAT1, may undergo similar trafficking, since almost nothing is known about their regulation.

## Regulatory differences along the intestinal axis

We observed significant differences in LAT4 food-entrained diurnal regulation along the longitudinal axis of the small intestine in WT mice (Figs 2–6). Duodenum was the only segment where the protein expression of LAT4 appeared to be regulated, being slightly increased during food anticipation and intake (ZT12-ZT16) (Fig 2A), coinciding with a decrease in pS274 (Fig 2B). In contrast, in jejunum and ileum only the phosphorylation of LAT4 showed food-entrained diurnal regulation with significant decrease at the level of S274 between ZT12 and ZT16. This effect was stronger and lasted longer in ileum (ZT8 to ZT20), reaching a near total pS274 absence (Figs 3–6). Such differential regulation coincides with recent discoveries in the field [6, 7], and highlights the heterogeneity of the small intestine.

Even though the majority of nutrients are absorbed in proximal jejunum [64], ileum receives both unabsorbed nutrients and nutrients from shed gastrointestinal epithelial cells, such that it might be important to keep ileal LAT4 at a substantial functional level to prevent amino acid wasting. As the last segment, ileum may also adjust its amino acid absorptive capacity by anticipation, based on information received about the luminal content of duodenum and jejunum and the plasma amino acid concentration.

## Regulation within insulin/mTORC1 pathway

Concurrently with the decrease in pS274 levels during the entrained food intake period (ZT16) (Figs 3–6), we also observed a strong pS6 staining in the ileal crypts (Fig 7F and 7G), similar to the effect that has been described in refed mice after overnight starvation [65]. Overall, our data show a strong underlying food-entrained diurnal rhythm of S6 phosphorylation in ileal villi (and trend in crypts), whilst in jejunum such a response was barely present (Table 3) (Fig 7). We also observed that in the isolated intestinal villi, both insulin and amino acid treatments were able to increase pS297 levels. In the case of insulin, this effect appeared to be sensitive to rapamycin—an inhibitor of mTORC1 (Fig 13). We thus hypothesize that additionally to the food-entrained diurnal regulation, LAT4 phosphorylation could be also regulated by the insulin/mTORC1 signaling pathway. This pathway might mediate the response to the actual intracellular amino acid concentration, which depends on the luminal and/or basolateral amino acid uptake and on intracellular protein degradation.

However, the most surprising observation was the strong decrease of ileal pS6 at ZT4 followed by an extensive increase at ZT8 in food-entrained mice (Fig 7E–7G). ZT8 is a time point during the resting phase, 12 h after food removal (Fig 1A). A possible explanation could be that the absence of food for 12 h led to a decrease in mTORC1 activity and subsequent activation of autophagy around ZT4 (Fig 7E–7G). This autophagy would lead to a transient re-increase in intracellular free amino acids with subsequent mTORC1 stimulation and phosphorylation of S6 [66, 67], that appears to be superposed to the underlying diurnal regulation, as shown by cosine wave fit to ileal pS6 values in villi when ZT8 is excluded (Table 3) (Fig 7E). The possibility of autophagy induction around ZT4 is supported by the recent observation that two key regulators of autophagy TFEB and TFE3 show diurnal rhythmicity with activation during the light, resting phase in mice [68].

### Dietary regulation

Our study showed that food entrainment of WT mice with low protein (8%) diet induced an increase in LAT4 intracellular localization in jejunum and also an increase in S274 phosphorylation in both jejunum and ileum during food anticipation, in comparison with mice entrained with standard (18%) or high (40%) protein diets (Figs 8–11). As insulin has been shown to be decreased in plasma of mice on low protein diets (3–6%)[69], it may be suggested that this hormone and its downstream effector mTORC1 that is also sensitive to amino acid concentration are involved in mediation of the observed LAT4 phosphorylation and subcellular localization changes. Interestingly, this increase in S274 phosphorylation under entrainment with low protein diet was not associated with a significant reduction of LAT4 protein expression, but rather with LAT4 internalization, such that a regulatory translocation of LAT4 to the basolateral membrane to increase transport capacity during food intake might still remain possible.

In contrast, increasing dietary protein concentration from 18% to 40% induced an increase in jejunal LAT4 protein expression without any changes in pS274 levels during entrained food anticipation (Fig 8C and 8D). It may be that under entrainment with standard 18% protein diet jejunal LAT4 expression and dephosphorylation on S274 was already close to its maximal level during food anticipation and thus no further increase was observed in 40% protein diet entrained mice. Interestingly, in ileum there was a trend for higher pS274 level and LAT4 intracellular localization under entrainment with high protein diet—similar, but weaker response than observed under low protein diet entrainment (Figs 10 and 11).

### Regulatory pathways involved in LAT4 phosphorylation revealed in *ex vivo* intestinal villi fractions

To investigate whether PKA and insulin could regulate LAT4, we used *ex vivo* intestinal villi fraction. To our knowledge, intestinal fractionation so far has been used mostly to isolate intestinal crypts to detect crypt-specific gene expression or to culture intestinal organoids [38, 39]. However, villi fractions turned out to be an excellent *ex vivo* model for testing the effect of diverse treatments on phosphorylation cascades in intestinal epithelial cells. Nonetheless, this model has also certain disadvantages. Isolated intestinal fractions lack the normal underlying intestinal structure, the nervous system and circulating factors. Enteric nerves have indeed been shown to be implicated in nutrient handling, glucose sensing and regulation of glucose transporter SGLT1 expression levels [48, 70–72] and thus might play a crucial role in communicating the presence and concentrations of nutrients and/or factors between the intestinal segments. Also, circulating humoral signals are known to play an important role in normal diurnal rhythm generation and adaptive regulation [28]. However, the absence of these

regulatory systems/pathways in villi fractions can also be an advantage, since it allows the testing of possible cell-autonomous regulation in response to specific humoral signals like insulin.

Using isolated intestinal villi fractions we discovered that the treatment with PKA activator forskolin led to a strong increase in phosphorylation of LAT4 on S274. This increase was sensitive to PKA inhibitor H89 and we observed a similar response also on other PKA substrates, confirming the involvement of PKA (Fig 12). We hypothesize that PKA phosphorylates S274 directly, as it lies within a typical corresponding motif [37]. However, the specific upstream signaling pathway/-s that activate PKA and lead to LAT4 phosphorylation changes in intestinal epithelial cells is/are still unknown to us. PKA has been shown to be involved in both the diurnal rhythm and dietary regulation. In the master circadian regulator—suprachiasmatic nuclei, PKA activation has been shown to slow down the circadian clock and to function as an important upstream effector of multiple circadian clock elements [73–75]. However, to our knowledge no published studies have yet investigated the role of PKA in the intestinal circadian/diurnal rhythmicity. Regarding the dietary regulation, most of the published studies have been done in yeast, where activation of PKA is essential in glucose sensing and uptake, and for the suppression of autophagy [76–78]. In contrast to the situation reported in yeast, PKA activity has been shown to be upregulated by glucose starvation in hepatic cells of mice [79]. Similar to its role in diurnal rhythmicity, the role of PKA in intestinal dietary regulation is yet to be investigated.

Overall, this study is the first proof that phosphorylation on a basolateral amino acid uniporter can be regulated within various metabolic pathways and can be under tight food-entrained diurnal and dietary control, suggesting its functional regulation on a post-translational level.

We also demonstrate the importance of studying all segments of the small intestine to reveal possible regulatory differences and to better describe their specific roles. Understanding how our body adapts to the daily rhythm and food content needs to include studies on nutrient transporters, and our study highlights how complex their adaptive regulation can be.

## Supporting information

**S1 Fig. In vivo treatment with ghrelin has no effect on LAT4 expression or phosphorylation.** Food-entrained mice received an injection with either pure saline or saline with 10 μg ghrelin and were sacrificed 10 min later. **A**: Concentration of growth hormone in mouse plasma determined the efficiency of ghrelin injections. **B**: LAT4 protein expression in jejunum. **C**: LAT4 phosphorylation on S274 in jejunum. **D**: LAT4 phosphorylation on S297 in jejunum. **B to D**: Total lysates from intestinal villi fraction of WT and LAT4 conditional KO mice analyzed by Western blot. For quantification, all values were normalized to beta actin. Further normalization to LAT4 ratio was done for phospho-specific antibodies. Quantification is shown in the left panel, representative blot in the right panel. Statistical analysis performed with One-way ANOVA using Tukey's multiple comparison test, *$p < 0.05$. Mean (SD), n = 6 mice per injection group from single experiment.
(TIF)

**S1 Raw Images. Original blot images.** Compilation of all original blot images. Panels that were used for corresponding figures as well as the additional modifications are indicated.
(PDF)

## Acknowledgments

We thank Zurich Integrative Rodent Physiology Facility and the Zurich Center for Microscopy and Image analysis for their help and equipment. We also thank the developer of Cosinor.

Online program Lubos Molcan for making the necessary adjustments to the program allowing us to perform our cosine wave fits.

## Author Contributions

**Conceptualization:** Lalita Oparija-Rogenmozere, Anuradha Rajendran, Nadège Poncet, Simone M. R. Camargo, François Verrey.

**Data curation:** Lalita Oparija-Rogenmozere, Anuradha Rajendran, Nadège Poncet.

**Formal analysis:** Lalita Oparija-Rogenmozere, Anuradha Rajendran.

**Funding acquisition:** François Verrey.

**Investigation:** Lalita Oparija-Rogenmozere, Anuradha Rajendran, Nadège Poncet, Simone M. R. Camargo.

**Methodology:** Lalita Oparija-Rogenmozere, Simone M. R. Camargo, François Verrey.

**Resources:** François Verrey.

**Supervision:** Nadège Poncet, Simone M. R. Camargo, François Verrey.

**Validation:** Lalita Oparija-Rogenmozere, Nadège Poncet, François Verrey.

**Visualization:** Lalita Oparija-Rogenmozere.

**Writing – original draft:** Lalita Oparija-Rogenmozere, François Verrey.

**Writing – review & editing:** Lalita Oparija-Rogenmozere, Anuradha Rajendran, Nadège Poncet, Simone M. R. Camargo, François Verrey.

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
