## [Decision Letter · Decision Letter 0]

3 Mar 2020

PONE-D-20-02951

Phosphorylation of mouse intestinal basolateral amino acid uniporter LAT4 is under circadian and dietary control

PLOS ONE

Dear Dr. Verrey,

Thank you for submitting your manuscript to PLOS ONE. After careful consideration, we feel that it has merit but does not fully meet PLOS ONE’s publication criteria as it currently stands. Therefore, we invite you to submit a revised version of the manuscript that addresses the points raised during the review process.

Please address the methodological and interpretational concerns raised by the reviewers. Specifically, add ad libitum fed animals as control group (ad libitum means that these animals have access to food over 24 hours). The data have been generated under diurnal conditions and not circadian conditions. Therefore, the conclusions should also be made under that aspect (the presence of a light/dark cycle). The experiment suggested by reviewer 1 under point 3 should be performed and added to the manuscript. Quantification of the data (point 4 reviewer 1 should also be included). 

We would appreciate receiving your revised manuscript by Apr 17 2020 11:59PM. To enhance the reproducibility of your results, we recommend that if applicable you deposit your laboratory protocols in protocols.io, where a protocol can be assigned its own identifier (DOI) such that it can be cited independently in the future. For instructions see: http://journals.plos.org/plosone/s/submission-guidelines#loc-laboratory-protocols

We look forward to receiving your revised manuscript.

Kind regards,

Urs Albrecht, Ph.D.

Academic Editor

PLOS ONE

Journal Requirements:

Reviewers' comments:

Reviewer's Responses to Questions

**Comments to the Author**

1. Is the manuscript technically sound, and do the data support the conclusions?

Reviewer #1: No

Reviewer #2: Yes

2. Has the statistical analysis been performed appropriately and rigorously? 

Reviewer #1: Yes

Reviewer #2: Yes

3. Have the authors made all data underlying the findings in their manuscript fully available?

Reviewer #1: Yes

Reviewer #2: Yes

4. Is the manuscript presented in an intelligible fashion and written in standard English?

Reviewer #1: Yes

Reviewer #2: Yes

5. Review Comments to the Author

Reviewer #1: Oparija-Rogenmozere and colleagues investigated the expression and phosphorylation of essential amino acid transporter Lat4 (Slc43a2) in three regions of the intestine of mice fed from early to late night. The results indicate differential expression and phosphorylated states of Lat4 according to intestine regions (i.e. duodenum, jejunum and ileum) and amount of proteins in the diet (8, 18 or 40%).

As detailed below, the manuscript raises several methodological and interpretative issues that need to be addressed.

1. Food-entrained versus ad lib-fed mice

It is not fully clear why all mice had food access limited to 8 h, starting at dark onset. Here food access is limited to the usual feeding period which might have enhanced daily rhythmicity. Why were mice fed ad libitum not studied as control group? Otherwise, what is the advantage or the purpose to study food-restricted mice fed only at night? Please specify.

2. Circadian control

Reading the present manuscript raises comments on the concept of circadian clocks. Circadian rhythms refer to endogenous biological rhyhms that are controlled by self-sustained clocks. Such endogenopus rhythms can only be assessed in constant environmental conditions (including constant light or dark, and constant ambient temperature). When a light-dark cycle is present, as it is the case in the present study, the parameters studied over 24 h should be called « daily » or « diurnal » rhythms. As a consequence, the word « circadian » such as used in the title is inappropriate for the present experiment and should be omitted throughout the manuscript.

3. Clock-controlled versus feeding-induced regulation

This study reports daily variations of several parameters such as protein levels of LAT4 in the duodenum and LAT4-pS297 in jejunum. When higher values are found at night, the present protocol does not allow discriminating between putative clock-controlled or feeding-induced regulation.

To test for feeding-induced control, another group of mice should be fed during daytime only (with the assumption that there will be a 12-h shift in the daily variations). The same remark holds for food-anticipatory mechanisms.

To test for clock-controlled regulation, another group of mice should be fasted (with the assumption that the daily variations will persist if they are regulated by endogenous clocks).

If ever the authors cannot add these new groups of mice, they should be much more cautious in their discussion on the interpretation of their findings, so as not to mislead the reader.

4. Image analysis and quantification

In figures 9 and 11, the authors consider that « low protein diet leads to increased intracellular localization of LAT4 in jejunum » and this diet « has no strong impact on LAT4 subcellular localization in ileum ». It seems that these respective statements are based on visual inspection only. Image analysis and quantification are required for both parameters.

5. Circadian clock in the intestine

It is surprising for a study supposed to investigate clock-related variations in the intestine that the authors did not cite previous investigations of circadian oscillations in that organ. Actually, the 2 proposed citations (one original work and one review) do not deal at all with the intestinal clock (i.e. Damiola et al. 2000 and Welsh et al. 2010, see refs 25 and 26, respectively).

Alternative suggestions:

Hepatic, duodenal, and colonic circadian clocks differ in their persistence under conditions of constant light and in their entrainment by restricted feeding. Polidarová L, Sládek M, Soták M, Pácha J, Sumová A. Chronobiol Int. 2011 Apr;28(3):204-15. doi: 10.3109/07420528.2010.548615.

PMID:21452916

Clock gene expression in the murine gastrointestinal tract: endogenous rhythmicity and effects of a feeding regimen. Hoogerwerf WA, Hellmich HL, Cornélissen G, Halberg F, Shahinian VB, Bostwick J, Savidge TC, Cassone VM. Gastroenterology. 2007 Oct;133(4):1250-60. Epub 2007 Jul 12.

PMID: 17919497

6. Miscellaneous

The main text indicates that mice were fed from ZT12 to ZT20, while Figure 1 suggests that mice had food access from ZT12 to ZT18. Please correct where needed.

« Rest » or « resting » period, rather than « passive » period.

Reviewer #2: Most research on circadian entrainment to feeding has focused on neural mechanisms and the liver, so it is quite interesting to learn about changes in the intestine in the study of Oparija-Rogenmozere and colleagues. They describe the expression of uni-, sim- and anti-porters in response to scheduled feeding and increased/decreased dietary protein content, eventually focusing their study on Lat4, an amino acid transporter that they have studied previously that is regulated by phosphorylation. In the present study they examine the expression and regulation of this transporter in great deal and the study appears technically sound in all ways.

I have several suggests for edits and clarification below:

Introduction, line 1, the word “laying” should be deleted

Most of the uses of the word “already” are unnecessary and it can be deleted without changing the meaning of the sentences

Introduction sentence: “Small intestine also possesses a self-sustaining circadian rhythm for which the most powerful entrainment is food intake (25, 26).” Reference 26 is not correct.

Introduction, last paragraph, please consider citing some of the studies that cast some doubt on the necessity of ghrelin peptide as the mechanism of behavioral anticipation of scheduled meals:

Gunapala et al https://doi.org/10.1371/journal.pone.0018377, which demonstrates that ghrelin deletion does not alter food anticipatory activity in mice

And also Daily et al 2012 doi: 10.1210/en.2011-1464, which dissociates gut peptide entrainment from behavioral entrainment

Also state whether that the ghrelin receptor is expressed in the intestine and that you were seeking to test for a direct action of ghrelin on the intestinal cells

Proofread by a native speaker--many sentences are missing the article “The”

Table I, replace “gender” with “sex”

Methods, was one week of inverted LD really enough to shift light entrained rhythms? Was this checked behaviorally?

Methods, It is not necessary to denote “JAX™” when describing C57BL/6J mice as the Jax is implied

Methods, “45oC”, remove “C”

Results, “Thus, the expression of LAT4 protein could be regulated at the level of translation or post-translationally (e.g. protein stability and/or translocation to membrane).”. This later bit about translocation should be removed as it doesn’t appear that the authors specifically isolated membrane fraction (versus other intracellular compartments) with the low speed centrifugation step that was applied. If more effort was taken to isolate membranes than what is described in the Methods section then it should be described there

What is the meaning of “Swiss roll”? Please define

There are at least two references to unpublished manuscripts. I believe that it is against the policy of PLoS ONE to cite unpublished manuscripts. At least that was the case the last time I published in PLoS ONE in 2018.

Link to phospho web domains needed only in methods section

For the “box and whisker” plots, it is only necessary to describe their meaning once instead of repeatedly in each figure legend.

Please attempt to fit a sine wave to the data that you believe to be showing a circadian rhythm and describe the goodness of fit to a sine wave. I appreciate that the data points are shown but it would be nice to see if these are rhythmic or not as many of them just appear noisy.

6. PLOS authors have the option to publish the peer review history of their article (what does this mean?). If published, this will include your full peer review and any attached files.

Reviewer #1: No

Reviewer #2: Yes: Andrew Steele

---

## [Author Response · Author response to Decision Letter 0]

6 Apr 2020

All responses to reviewers and editor can be found in the submitted file "Response to Reviewers"

---

## [Decision Letter · Decision Letter 1]

27 Apr 2020

PONE-D-20-02951R1

Phosphorylation of mouse intestinal basolateral amino acid uniporter LAT4 is controlled by food-entrained diurnal rhythm and dietary proteins

PLOS ONE

Dear Dr. Verrey,

Thank you for submitting your manuscript to PLOS ONE. After careful consideration, we feel that it has merit but does not fully meet PLOS ONE’s publication criteria as it currently stands. Therefore, we invite you to submit a revised version of the manuscript that addresses the points raised during the review process.

We would appreciate receiving your revised manuscript by Jun 11 2020 11:59PM. To enhance the reproducibility of your results, we recommend that if applicable you deposit your laboratory protocols in protocols.io, where a protocol can be assigned its own identifier (DOI) such that it can be cited independently in the future. For instructions see: http://journals.plos.org/plosone/s/submission-guidelines#loc-laboratory-protocols

We look forward to receiving your revised manuscript.

Kind regards,

Urs Albrecht, Ph.D.

Academic Editor

PLOS ONE

Reviewers' comments:

Reviewer's Responses to Questions

**Comments to the Author**

1. If the authors have adequately addressed your comments raised in a previous round of review and you feel that this manuscript is now acceptable for publication, you may indicate that here to bypass the “Comments to the Author” section, enter your conflict of interest statement in the “Confidential to Editor” section, and submit your "Accept" recommendation.

Reviewer #1: All comments have been addressed

Reviewer #2: All comments have been addressed

2. Is the manuscript technically sound, and do the data support the conclusions?

Reviewer #1: Yes

Reviewer #2: (No Response)

3. Has the statistical analysis been performed appropriately and rigorously? 

Reviewer #1: Yes

Reviewer #2: (No Response)

4. Have the authors made all data underlying the findings in their manuscript fully available?

Reviewer #1: (No Response)

Reviewer #2: (No Response)

5. Is the manuscript presented in an intelligible fashion and written in standard English?

Reviewer #1: Yes

Reviewer #2: (No Response)

6. Review Comments to the Author

Reviewer #1: (No Response)

Reviewer #2: Excellent job on revising the manuscript. I'm very sorry that I did not mention this upon my first review, but the style that is used in parts of the results section seems inappropriate. For example, parts of the results section read like Figure legends in stead of as a narrative that references Figures parenthetically to support the description.

This is a copy-past of an early paragraph within the results section:

Fig 1. mRNAs of nutrient uniporters in mice duodenum show little or no food 504

entrained diurnal regulation, in contrast to symporters and antiporters. A: Food505

entrainment regimen used for all mice experiments. ZT0: start of the resting period; ZT12:

506 start of the active period. Mice fed 18% protein diet only from ZT12 to ZT20 for at least

507 14 consecutive days. Euthanasia at ZT12 was done before the feeding starts. B: mRNA

508 expression of different nutrient transporters in duodenum measured by real time PCR.

One cannot write a results section in this manner--this should be in the figure legends and the results should be a narrative written with proper sentences in paragraph form. For example, "The mRNA of nutrient uniporters in the duodenum of mice showed little or no food entrained diurnal regulation, in contrast to x (Figure 1A)... (Figure 1B) and so on.

7. PLOS authors have the option to publish the peer review history of their article (what does this mean?). If published, this will include your full peer review and any attached files.

Reviewer #1: No

Reviewer #2: Yes: Andrew Steele

---

## [Author Response · Author response to Decision Letter 1]

1 May 2020

For responses please view the updated Cover letter and the Responses for Reviewers file.

---

## [Editor Report · Decision Letter 2]

11 May 2020

PONE-D-20-02951R2

Phosphorylation of mouse intestinal basolateral amino acid uniporter LAT4 is controlled by food-entrained diurnal rhythm and dietary proteins

PLOS ONE

Dear Dr. Verrey,

Thank you for submitting your manuscript to PLOS ONE. After careful consideration, we feel that it has merit but does not fully meet PLOS ONE’s publication criteria as it currently stands. Therefore, we invite you to submit a revised version of the manuscript that addresses the points raised during the review process.

We would appreciate receiving your revised manuscript by Jun 21 2020 11:59PM. To enhance the reproducibility of your results, we recommend that if applicable you deposit your laboratory protocols in protocols.io, where a protocol can be assigned its own identifier (DOI) such that it can be cited independently in the future. For instructions see: http://journals.plos.org/plosone/s/submission-guidelines#loc-laboratory-protocols

We look forward to receiving your revised manuscript.

Kind regards,

Urs Albrecht, Ph.D.

Academic Editor

PLOS ONE

Additional academic editor comments;

Reviewer two had a minor comment on the R1, and this second new comment has to be added to the revision. Both reviewers have been invited to look at R1, the first one accepted it and the second one had a minor comment. These are the standings. So the authors should correct the second comment of reviewer 2.

---

## [Author Response · Author response to Decision Letter 2]

13 May 2020

Rebuttal letter for PONE-D-20-02951R2 decision

Dear Academic Editor, dear Dr Albrecht

thank you for your message that confirms that there is only a little editorial misunderstanding left regarding the minor comment of Reviewer #2. 

We had indeed addressed (rebutted) this minor comment in our letter of May 1st, explaining that the problem raised by reviewer #2 was due to the fact that he had not realized that we had inserted the Figure legends within the text at the bottom of the corresponding paragraphs to comply with the editorial policy of PLOS ONE and therefore he had considered that the style of the text (actually the Legends) was not appropriate for the Result section. 

You can indeed read in point 6 'Review Comments to the Author' of the 'PLOS ONE Decision: Revision required [PONE-D-20-02951R1]' mail of April 27 the additional point of Reviewer #2 that includes a pasted version of our Legend for Figure 1 and a suggestion how to use a narrative style in the Result section. 

We believe however that the style of our Result section text (that does not include the Figure legends) is indeed written in a narrative style with proper sentences.

Therefore, we resubmit the identical manuscript and hope that after this explanation, you will consider that it fulfills the criteria for publication in PLOS ONE.

---

## [Editor Report · Decision Letter 3]

14 May 2020

Phosphorylation of mouse intestinal basolateral amino acid uniporter LAT4 is controlled by food-entrained diurnal rhythm and dietary proteins

PONE-D-20-02951R3

Dear Dr. Verrey,

We are pleased to inform you that your manuscript has been judged scientifically suitable for publication and will be formally accepted for publication once it complies with all outstanding technical requirements.

With kind regards,

Urs Albrecht, Ph.D.

Academic Editor

PLOS ONE

Additional Editor Comments (optional):

Thank you for your revisions and resolve the misunderstanding.
---

## [Editor Report · Acceptance letter]

21 May 2020

PONE-D-20-02951R3 

Phosphorylation of mouse intestinal basolateral amino acid uniporter LAT4 is controlled by food-entrained diurnal rhythm and dietary proteins 

Dear Dr. Verrey:

I am pleased to inform you that your manuscript has been deemed suitable for publication in PLOS ONE. Congratulations! Your manuscript is now with our production department. 

With kind regards,

on behalf of

Prof. Urs Albrecht 

Academic Editor

PLOS ONE